



# Retrieving ice nucleating particle concentration and ice multiplication factors using active remote sensing validated by in situ observations

Jörg Wieder[1], Nikola Ihn[1], Claudia Mignani[2], Moritz Haarig[3], Johannes Bühl[3], Patric Seifert[3], Ronny Engelmann[3], Fabiola Ramelli[1], Zamin A. Kanji[1], Ulrike Lohmann[1], and Jan Henneberger[1]

[1]ETH Zurich, Institute for Atmospheric and Climate Science, Zurich, Switzerland
[2]Department of Environmental Sciences, University of Basel, Basel, Switzerland
[3]Leibniz Institute for Tropospheric Research, Leipzig, Germany

**Correspondence:** Jörg Wieder (joerg.wieder@env.ethz.ch), and Jan Henneberger (jan.henneberger@env.ethz.ch)

**Abstract.** Understanding the evolution of the ice phase within mixed-phase clouds (MPCs) is necessary to reduce uncertainties related to the cloud radiative feedback in climate projections and precipitation initiation. Both primary ice formation via ice nucleating particles (INPs) and secondary ice production (SIP) within MPCs are unconstrained, not least because of the lack of atmospheric observations. In the past decades, advanced remote sensing methods have emerged which provide high resolution

data of aerosol and cloud properties and could be key in understanding microphysical processes on a global scale. In this study, we retrieved INP concentrations, and ice multiplication factors (IMFs) in wintertime orographic clouds using active remote sensing and in situ observations obtained during the RACLETS campaign in the Swiss Alps. INP concentrations in air masses dominated by Saharan dust and continental aerosol were retrieved from a polarization Raman lidar and validated with aerosol and INP in situ observations on a mountaintop. A calibration factor of 0.0204 for the global INP parameterization by DeMott

et al. (2010) is derived by comparing in situ aerosol and INP measurements improving the INP concentration retrieval for continental aerosols. Based on combined lidar and radar measurements, the ice crystal number concentration and ice water content were retrieved and validated with balloon-borne in situ observations, which agreed with the balloon-borne in situ observations within an order of magnitude. For seven cloud cases the ice multiplication factors (IMFs), defined as the quotient of the ice crystal number concentration to the INP concentration, were calculated. The median IMF was around 80 and SIP

was active (defined as IMFs > 1) nearly 85% of the time. SIP was found to be active at all observed temperatures ($-30\,°C$ to $-5\,°C$) with highest IMFs between $-20\,°C$ and $-5\,°C$. The introduced methodology could be extended to larger datasets to better understand the impact of SIP not only over the Alps but also at other locations and for other cloud types.

## 1   Introduction and background

The increase of the Earth's global mean temperature in recent years is unequivocal, yet the extent of a cloud cooling effect

remains most uncertain (IPCC, 2021). The radiative feedback of a cloud is a strong function of the hydrometeor phase (Sun and Shine, 1994). Mixed-phase clouds (MPCs) consisting of both water phases (liquid and solid) contribute strongly to uncer-





tainties in the radiative feedback. They are thermodynamically unstable (see e.g., Korolev et al., 2017) because of the lower vapor pressure with respect to ice than with respect to liquid water. This causes the ice crystals to grow at the expense of the evaporation of cloud droplets which is referred to as the Wegener-Bergeron-Findeisen process (Wegener, 1911; Bergeron, 1935; Findeisen, 1938). The complexity of phase partitioning adds to difficulties simulating MPCs in models (e.g., McCoy et al., 2016; Matus and L'Ecuyer, 2017) where experimental observations could reduce uncertainties (see e.g., Baumgardner et al., 2011; Mahrt et al., 2019). In the evolution of a MPC, the ice phase plays an important role as it controls precipitation initiation and consequently cloud life time (see e.g., Field and Heymsfield, 2015; Heymsfield et al., 2020). Thus, the ice phase does not only determine how a cloud impacts the radiative budget but also for how long. The first ice crystals in a cloud can be either formed within the cloud by homogeneous or heterogeneous ice nucleation or can be externally introduced by e.g., sedimentation from a higher cloud (*seeder-feeder process*, e.g., Proske et al., 2021; Ramelli et al., 2021a, and references therein), or levitated from the ground (*blowing snow*, especially in mountainous terrain, e.g., Beck et al., 2018; Walter et al., 2020, and references therein). Approximately below $-38\,°C$, supercooled droplets can freeze homogeneously or heterogeneously. Above that temperature, heterogeneous ice nucleation is favored on sparsely abundant aerosols called ice nucleating particles (INPs, see e.g.,Wegener (1911); Vali (1971); Pruppacher and Klett (2010); Murray et al. (2012)). Until today, a variety of aerosol particles acting as INPs in the atmosphere have been identified such as desert and soil dust, organics from biomass burning, marine or terrestrial biogenic particles, atmospherically aged soot, bacteria, and others (Kanji et al., 2017; Huang et al., 2021). However, due to the geospatial variability of INP sources, atmospheric INP concentrations feature a high spatiotemporal variability complicating their quantitative assessment (see e.g., DeMott et al., 2010; Kanji et al., 2017; Murray et al., 2021, and references therein). Understanding heterogeneous ice formation and subsequent ice crystal growth is key to understanding the link between aerosols and precipitation formation, and is therefore an important step towards constraining weather and climate predictions (Ansmann et al., 2019a; Bühl et al., 2019). After first ice crystals are found in a cloud, secondary ice production (SIP) can enhance the ice crystal number concentration (ICNC) by various processes, e.g., fragmentation during ice-ice collisions (e.g., Vardiman, 1978; Takahashi et al., 1995), or splintering during riming (*Hallett-Mossop process*, active between $-8\,°C$ and $-3\,°C$, Hallett and Mossop, 1974), given favorable environmental conditions (Korolev and Leisner, 2020). The Hallet-Mossop process requires the presence of supercooled liquid droplets. Thus, the Hallett-Mossop process can play an important role especially over orographic terrain, given that the orographic forcing induced updrafts could sustain the availability of liquid water droplets (Lohmann et al., 2016). Various field studies have observed ICNC exceeding the ambient INP concentration by several orders of magnitude (see e.g., Koenig, 1963; Auer et al., 1969; Hobbs and Rangno, 1985, 1990, 1998; Gayet et al., 2009; Crosier et al., 2011; Stith et al., 2011; Crawford et al., 2012; Heymsfield and Willis, 2014; Lawson et al., 2015; Lasher-Trapp et al., 2016; Ladino et al., 2017; Mignani et al., 2019; Lauber et al., 2021; Pasquier et al., 2021). However, the underlying SIP processes are weakly constrained (Korolev and Leisner, 2020). Therefore, the prevalence of SIP in the atmosphere remains uncertain as well as the environmental conditions for SIP to be active.

Uncertainties in predicting atmospheric INP concentrations and ice multiplication are (partly) related to spatiotemporally limited field observations, because of the large effort needed to obtain field data and their point-like characteristic. Remote sensing techniques can be a suitable solution to overcome the issue by providing continuous data in time and at least one



spatial dimension. Starting in the 1930s, remote sensing techniques have evolved to very advanced and reliable instruments today (Wandinger, 2005). Using lidar (light detection and ranging) and radar (radio detection and ranging) instruments, a broad variety of aerosol and cloud properties can be retrieved (e.g., Ansmann et al., 2019a). Lidar systems are key instruments

for the investigation of aerosol optical properties and their vertical layering in the atmosphere. The retrieved backscatter and extinction coefficients allow for the determination of physical properties of aerosol particles in the atmosphere, such as size and particle number concentration (see e.g., Müller et al., 1999; Veselovskii et al., 2005; Mamouri and Ansmann, 2016). Advanced lidar systems also deliver other products such as extinction-to-backscatter ratio (lidar ratio), linear depolarization ratio (LDR) or Ångström exponent from multiwavelength observations of the backscatter and extinction coefficients that allow

to determine more detailed aerosol information, e.g., aerosol type (see e.g., Burton et al., 2012; Groß et al., 2013; Baars et al., 2017). The accuracy of lidar retrievals has been validated in many studies with focus on different properties comparing lidar measurements with in situ observations using aircrafts or unmanned aerial vehicles (UAVs) (Ferrare et al., 1998; Wandinger et al., 2002; Sakai et al., 2003; Müller et al., 2012; Sawamura et al., 2017; Schrod et al., 2017; Marinou et al., 2019; Haarig et al., 2019; Düsing et al., 2021). Furthermore, the LDR was utilized to determine the contribution of dust to the observed

aerosol (Shimizu et al., 2004; Tesche et al., 2009; Mamouri and Ansmann, 2016; Haarig et al., 2017). Precise knowledge about the aerosol type is essential to use the lidar retrieval to estimate INP concentrations, owed to the fact that the INP concentration is deduced only from a physical property such as particle number concentration and surface area concentration. However, the parameterizations are often proposed for specific aerosol types such as dust (e.g., Niemand et al., 2012; DeMott et al., 2015; Ullrich et al., 2017; Harrison et al., 2019), marine aerosol (e.g., McCluskey et al., 2018), soot (e.g., Ullrich et al., 2017),

and global aerosol (e.g., DeMott et al., 2010). The feasibility of predicting INP concentration from lidar profiles has been shown by Mamouri and Ansmann (2016). In lidar INP studies, typically three aerosol type categories are used: mineral dust (Ansmann et al., 2003; Mamouri and Ansmann, 2016; Schrod et al., 2017; Haarig et al., 2019; Marinou et al., 2019), marine aerosol (Mamouri and Ansmann, 2016; Haarig et al., 2019), and continental aerosol (Mamouri and Ansmann, 2016; Schrod et al., 2017; Düsing et al., 2018; Marinou et al., 2019; Düsing et al., 2021). If necessary, these categories can be extended by

wildfire smoke (Ansmann et al., 2021; Engelmann et al., 2021) or volcanic ash (Ansmann et al., 2011). Whereas the aerosol sources within the first two categories can be comparably narrowed down by source, continental aerosol sources are much more diverse, featuring (among others) biological, organic, or lifted soil particles, and also anthropogenic emitted particles from e.g., biomass burning or combustion (Kanji et al., 2017; Huang et al., 2021, and references therein) complicating the retrieval of continental INP concentrations. Thus, finding the most ideal INP parameterization for a given aerosol type is key

for minimizing uncertainties in predicted INP concentration. Schrod et al. (2017) proposed calibration factors to optimize the INP parameterization by DeMott et al. (2015) for the retrieval of INP concentration in condensation and deposition mode from dust-dominated air masses. Marinou et al. (2019) showed that separating the dust-carrying air masses into a dust and a continental component and applying the parameterizations of DeMott et al. (2015) and DeMott et al. (2010), respectively, yields best agreement between lidar retrieval and in situ observations (basing on Schrod et al., 2017). However, Marinou et al.

(2019) stated further that *"additional measurements are required in order to define the optimum INP parameterizations for*





*nondust atmospheric conditions (e.g., continental, marine, smoke)"* suggesting that an optimal INP parameterization for the retrieval of INP concentration from continental air masses is yet to be proposed.

Remote sensing instrumentation has recently increasingly been used for cloud observations, and is becoming a promising method for retrieving ice crystal related parameters (see e.g., Seifert et al., 2010; Bühl et al., 2013; Zhang et al., 2014; Ansmann
et al., 2019a). It allows for a continuous monitoring of the vertical cloud structure, and therefore permits the observation of spatiotemporal cloud evolution (Seifert et al., 2010). This is a crucial addition to in situ measurements, which only allow for measurements in a constrained time and height frame (Seifert et al., 2010). The ice water content (IWC) can be retrieved solely based on radar measurements (Hogan et al., 2006) and ICNC can be estimated by linking the IWC to an assumed particle shape and size distribution (see e.g., Bühl et al., 2019). Complementing radar retrievals with size information from a collocated lidar
improves the accuracy of cloud property retrieval further (Delanoë et al., 2013). In previous studies, remote sensing was used to study (primary) ice nucleation in the atmosphere (see e.g., Sakai et al., 2003; Ansmann et al., 2008; Eidhammer et al., 2010; Seifert et al., 2010, 2011; Ansmann et al., 2019a; Engelmann et al., 2021) and to estimate SIP (see e.g., Auer et al., 1969; Luke et al., 2021; Sotiropoulou et al., 2021). The identification of a specific SIP process from atmospheric in situ and remote sensing observations is a challenge. Lauber et al. (2021) recently showed that recirculation of melted ice crystals can enhance
the concentration of small ice crystals in the proximity above the melting layer. Whereas deepening the understanding of individual processes seems best achievable in controlled conditions during laboratory studies (Korolev and Leisner, 2020), the contribution and magnitude of SIP still needs to be assessed in the (complex interacting) atmosphere. From field observations, the ice multiplication factor (IMF) defined as the ICNC divided by the INP concentration can be utilized to quantify SIP. Whereas the IMF allows to quantify the excess of ice crystals, it cannot be used to directly identify a specific SIP process due
to the potential (temporal and spatial) difference between the origin of the first ice crystals and SIP to occur.

In this study we combined a suite of in situ and remote sensing instruments to understand cloud formation and evolution in orographic terrain. In February and March 2019, we performed an intensive field campaign in the Swiss Alps in the region of Davos. In a high valley, a combined lidar-radar system was employed along with ground-based in situ aerosol (including INP) observation (Wieder et al., 2021) and balloon-borne in situ cloud observations (Ramelli et al., 2021a, b). A second in
situ aerosol site was located on a nearby mountaintop (height difference 1.1 km, Mignani et al., 2021; Wieder et al., 2021). The near collocation of the lidar beam and the mountaintop site (horizontal displacement 3.65 km) is ideally suited for aerosol and thus INP closure as also previously shown by Bedoya-Velásquez et al. (2018) which studied the hygroscopicity of aerosol particles. Based on aerosol data collected over eight weeks and seven observed cloud events, we address the following: First, we validate the lidar retrieval of aerosol number concentration and surface area concentration which is the basis for the INP
concentration retrieval (Section 3.1). Second, based on INP concentrations measured during a Saharan dust event, we assess the accuracy of different dust parameterizations and evaluate their performance (Section 3.2.1). Third, we evaluate two INP parameterizations (DeMott et al., 2010; Ullrich et al., 2017) used in the lidar community for the retrieval of INP concentration from continental air masses (Section 3.2.2). Based on our in situ measurements, we propose a calibration factor to optimize one INP parameterization and validate the tuning with the lidar observations (Section 3.2.3). Fourth, we validate radar retrieved





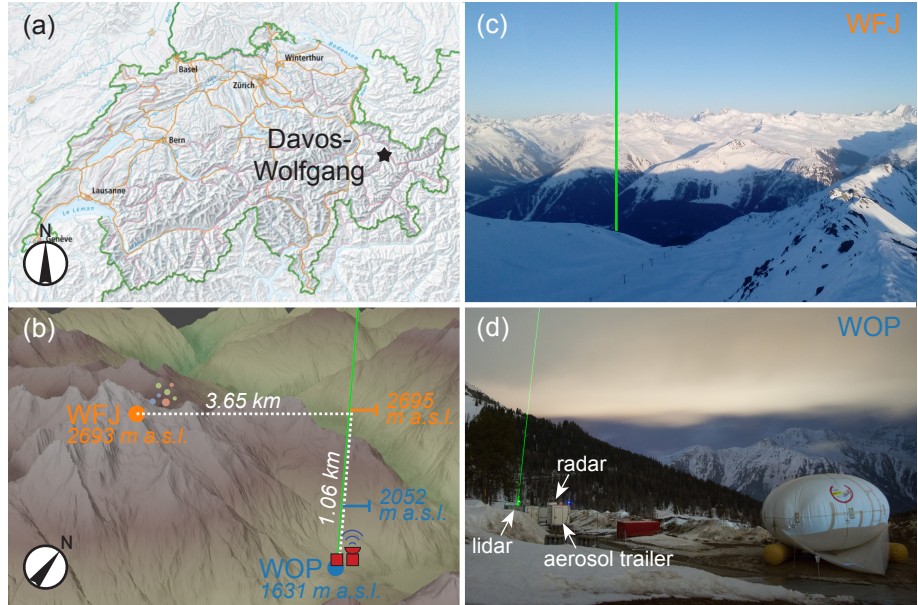

**Figure 1.** Overview of the measurement locations: (a) Davos-Wolfgang (black star) in the east of Switzerland (map source: Federal Office of Topography). (b) Measurement locations with the local topography around. The OceaNet container and the lower aerosol monitoring site were located at Wolfgangpass (blue dot, WOP, 1631 m a.s.l.). A second aerosol measurement site was located on Weissfluhjoch (orange dot, WFJ, 2693 m a.s.l.) which is located approximately 1.1 km higher than Wolfgangpass and horizontally displaced by 3.65 km. Lidar product retrieval heights for the comparison with in situ measurements at WFJ and WOP are indicated in orange and blue, respectively, right of the lidar beam (green). The topography was extracted from the digital height model DHM200 from the Federal Office of Topography swisstopo. (c) View from WFJ in direction of WOP. The approximate location of the lidar beam is indicated in green. (d) Placement of the aerosol trailer right of the OCEANET container's lidar and radar at WOP. The tethered balloon used for in situ cloud observations is seen in the front right.

IWC and ICNC with balloon-borne in situ observations (Section 3.3). Lastly, using the tuned parameterization we present a methodology to obtain estimates of IMFs in MPCs (Section 3.4).

## 2   Measurement setup and methodology

The RACLETS (Role of Aerosols and CLouds Enhanced by Topography on Snow) campaign took place in the region of Davos, Switzerland (Figure 1a), in February and March 2019, where an extensive set of aerosol, cloud, precipitation and snow
measurements were conducted (Envidat, 2019; Walter et al., 2020; Mignani et al., 2021; Ramelli et al., 2021a, b; Lauber et al., 2021; Georgakaki et al., 2021; Wieder et al., 2021). The two main measurement sites were located on a saddle at the entrance of a high valley (Wolfgangpass, 1631 m a.s.l., hereafter referred to as WOP) and the other on a mountaintop (Weissfluhjoch, 2693 m a.s.l., hereafter referred to as WFJ) as shown in Figure 1b. At both locations, two similarly equipped aerosol measurement sites were set up. In addition, the remote sensing instrumentation was installed at WOP.





## 2.1 In situ aerosol and INP measurements

The aerosol measurement sites at WFJ and WOP were previously presented Wieder et al. (2021); Georgakaki et al. (2021); Mignani et al. (2021). At both sites, ambient air was sampled through a 46°C-heated inlet. The degradation of relevant INPs (mostly biological) is unlikely as discussed in Wieder et al. (2021). Downstream, an Aerodynamic Particle Sizer Spectrometer (APS; Model 3321, TSI Inc., US) and a Scanning Mobility Particle Sizer Spectrometer (APS; Model 3938, TSI Inc., US) recorded aerosol size distributions between approximately 10 nm and 20 µm. In this study, aerodynamic diameter and electronic mobility were converted to physical diameter assuming a shape factor $\chi = 1.2$ and an assumed particle density $\rho = 2 \, \mathrm{g \, cm^{-3}}$ (Thomas and Charvet, 2017). Ambient aerosol was collected over a time span of 20 minutes into pure water (15 mL, W4502-1L, Sigma-Aldrich, US) for offline INP analysis using high flow-rate impingers (Coriolis® µ, Bertin Instruments, France, 300 L min$^{-1}$) attached to the end of the inlets (Wieder et al., 2021; Mignani et al., 2021). The (immersion mode) INP analysis was done on site utilizing the drop-freezing apparatuses LINDA (Stopelli et al., 2014) at WFJ and DRINCZ (David et al., 2019) at WOP. Further details about the aerosol setups as well as the INP data processing can be found in Wieder et al. (2021).

## 2.2 Lidar and radar measurements

During RACLETS, a 35-GHz (Ka-band) cloud radar and the multi-wavelength polarization Raman lidar Polly$^{\mathrm{XT}}$-OCEANET (Engelmann et al., 2016, hereafter referred to as Polly$^{\mathrm{XT}}$) of the Leibniz Institute for Tropospheric research (TROPOS) were deployed at the WOP site (Fig. 1d) as part of the mobile observation platform OCEANET-Atmosphere (Griesche et al., 2020). The instruments provided a continuous overview on the temporal evolution of the vertical distribution of aerosols and clouds from 8 February to 16 March 2019 and are used in the following for remote sensing based retrievals of aerosol and cloud microphysical properties.

Polly$^{\mathrm{XT}}$ measured vertical profiles of the particle backscatter coefficient (at 355, 532 and 1064 nm), the particle extinction coefficient with the Raman method (nighttime, 355 and 532 nm), and the particle LDR (355 and 532 nm). Based on the Polly$^{\mathrm{XT}}$ observations, INP number concentration was calculated following the POLIPHON method as described in Mamouri and Ansmann (2016). Here, only a short description of POLIPHON will be provided. In a first step, the dust and non-dust contribution to the backscatter coefficient are separated using the particle LDR at 532 nm. The backscatter contributions are transferred to extinction contributions using a lidar ratio of 55 sr for dust and 50 sr for continental particles (rural background, pollution). These extinction coefficients are converted to number concentrations of particles with radius larger than 250 nm ($n_{250}$) and surface area concentrations ($s$). The needed conversion factors are aerosol type dependent and are obtained from long-term sun-photometer (AERONET) data sets (Mamouri and Ansmann, 2016; Ansmann et al., 2019b). In this study, we apply extinction-to-number-concentration (particle radius > 250 nm) conversion factors of 0.19 Mm cm$^{-3}$ for Saharan dust and 0.10 Mm cm$^{-3}$ for continental aerosol and extinction-to-surface-area conversion factors of $2.4 \cdot 10^{-12}$ Mm m$^2$ cm$^{-3}$ for Saharan dust and $2.8 \cdot 10^{-12}$ Mm m$^2$ cm$^{-3}$ for continental aerosol (Mamouri and Ansmann, 2016; Ansmann et al., 2019b). With this method, the vertical profiles of the basic input parameters in various INP parametrizations are derived (see Sect. 3.2).



The 35-GHz cloud radar was of type Mira-35 (Görsdorf et al., 2015). During RACLETS, the radar was operated in vertical-stare mode. Pulses with a length of 208 ns were emitted at a repetition frequency of 6000 Hz, resulting in a vertical resolution of 31.17 m and a maximum unambiguous velocity range of $25.6 \, \mathrm{m \, s^{-1}}$, which spans from -12.8 to $12.8 \, \mathrm{m \, s^{-1}}$. The return signals of the emitted linearly polarized pulses were detected separately in the co- and cross-polarized planes. For both channels, Doppler spectra are derived from Fourier transformations of the return signals from a series of 512 consecutive pulses, corresponding to a Doppler-velocity resolution of $0.05 \, \mathrm{m \, s^{-1}}$. The final temporal resolution of the acquired cloud radar dataset of 10 s is obtained from incoherent averaging of 100 consecutive Doppler spectra.

ICNCs were retrieved from the cloud radar observations with the method described in Bühl et al. (2019). The procedure to derive ICNC from the RACLETS cloud radar observations was described by Ramelli et al. (2021a) and is in the following only explained briefly. Based on the observations of particle shape (LDR), radar reflectivity, Doppler velocity, and Doppler spectral width, the best fit to those values derived from simulated ice crystal number size distributions from a pre-calculated lookup table was obtained. The particle diameter was estimated from the particle terminal fall velocity and spectral width measured with the cloud radar.

## 2.3 In situ cloud measurements

During the campaign, cloud properties were also measured in situ using the tethered balloon system HoloBalloon (Ramelli et al., 2021a, b). On HoloBalloon, a HOLographic Imager for Microscopic Objects (HOLIMO) is installed which measures phase-resolved cloud properties (Ramelli et al., 2020). In a comparison to radar observations, we use IWCs and ICNCs measured with HoloBalloon on 8 March 2019 of which further details on the measurements and the synoptic situation are provided in Ramelli et al. (2021a).

## 2.4 Cloud temperature data

Temperatures at cloud top and within the clouds were retrieved from the COSMO-1 analysis data at corresponding heights above WOP. The COSMO-1 data was provided by MeteoSchweiz for the processing of the remote sensing data.

## 2.5 Investigated INP parameterizations

INP concentrations can be obtained from remote sensing data by applying a suitable INP parameterization on retrieved aerosol properties. By now, a wide range of INP parameterizations specific to a certain aerosol type (e.g., dust, continental, marine, soot) and a freezing mode (e.g., immersion or deposition) exist. Here, we compare lidar estimated INP concentrations with in situ observations in immersion mode at temperatures $\geq -20 \, °\mathrm{C}$ during times of a Saharan dust event and otherwise continental aerosol over the region of Davos. In Table 1, we present the existing parameterizations for immersion freezing which are applicable to air masses during our observations. The overview is based on the summary of Marinou et al. (2019) which gives detailed information about the parameterizations (Marinou et al., 2019, , Section 2). In the presence of Saharan dust-carrying air masses, we investigate the predictions from the parameterizations of DeMott et al. (2015) (D15), Ullrich et al.


**Table 1.** Overview of INP parameterizations for immersion freezing mode considered in this study along with the aerosol type they are applicable to, the aerosol property they are based on ($n_{250}$ : aerosol number concentration of particles with radius larger 250 nm, $s$ : aerosol surface area concentration), and the applicable temperature range. This overview was adapted from the more extensive summary given by Marinou et al. (2019, Table 1).

| Short name | Reference | Aerosol type | Aerosol property | Temperature range |
|---|---|---|---|---|
| D15 | DeMott et al. (2015) | dust | $n_{250}$ | $-35\,°C$ to $-21\,°C$ |
| U17d | Ullrich et al. (2017) | dust | $s$ | $-30\,°C$ to $-14\,°C$ |
| H19 | Harrison et al. (2019) | dust | $s$ | $-37.5\,°C$ to $-3.5\,°C$ |
| D10 | DeMott et al. (2010) | mixed* | $n_{250}$ | $-35\,°C$ to $-9\,°C$ |
| U17s | Ullrich et al. (2017) | soot | $s$ | $-34\,°C$ to $-18\,°C$ |

*Note that D10 was not primarily developed for predicting continental INP concentrations and includes samples from different field observations in North America, the Amazonian and the Pacific also featuring dust-carrying air masses. It was shown in recent publications (Mamouri and Ansmann, 2016; Marinou et al., 2019) that it is suitable to deduce continental INP concentration from lidar observations.

(2017) (dust parameterization, immersion freezing mode, U17d), and Harrison et al. (2019) (H19). The latter is a more recent dust parameterization scaling with the relative contribution of K-feldspar which was found to correlate strongest with the INP

concentration in their study. In the other cases, which are considered as air masses carrying continental aerosol, we investigate the INP concentration prediction based on DeMott et al. (2010) (global parameterization, D10) and Ullrich et al. (2017) (soot parameterization, immersion freezing mode, U17s). The latter was developed on soot aerosol, thus, making it suitable to specifically capture anthropogenic contributions to continental aerosol. Notably, the application ranges of all parameterizations presented in Table 1 are applicable mainly at temperature $\leq -15\,°C$. Therefore, we extrapolate the parameterizations to $-5$

$°C$ in accordance with Marinou et al. (2019).

## 3 Results and discussion

Atmospheric INP concentrations and IMF are estimated from the lidar and radar measurements. Three main uncertainties are involved in the estimation: (i) uncertainties linked to the measurement of the aerosol extinction coefficient and its conversion to number or surface area concentration, (ii) uncertainties linked to the INP parameterization itself, and (iii) using an param-

eterization not suitable for the dominant aerosol constituent. In the following, we first validate the aerosol properties as input and consequently determine the most suitable INP parameterization for dust and continental aerosol during RACLETS.

### 3.1 Aerosol concentration comparison

In the following, we investigate the accuracy of the lidar retrieval of aerosol properties with in situ observations. For the comparison of lidar observations to in situ observations at WFJ and WOP, lidar retrievals were taken from the closest height



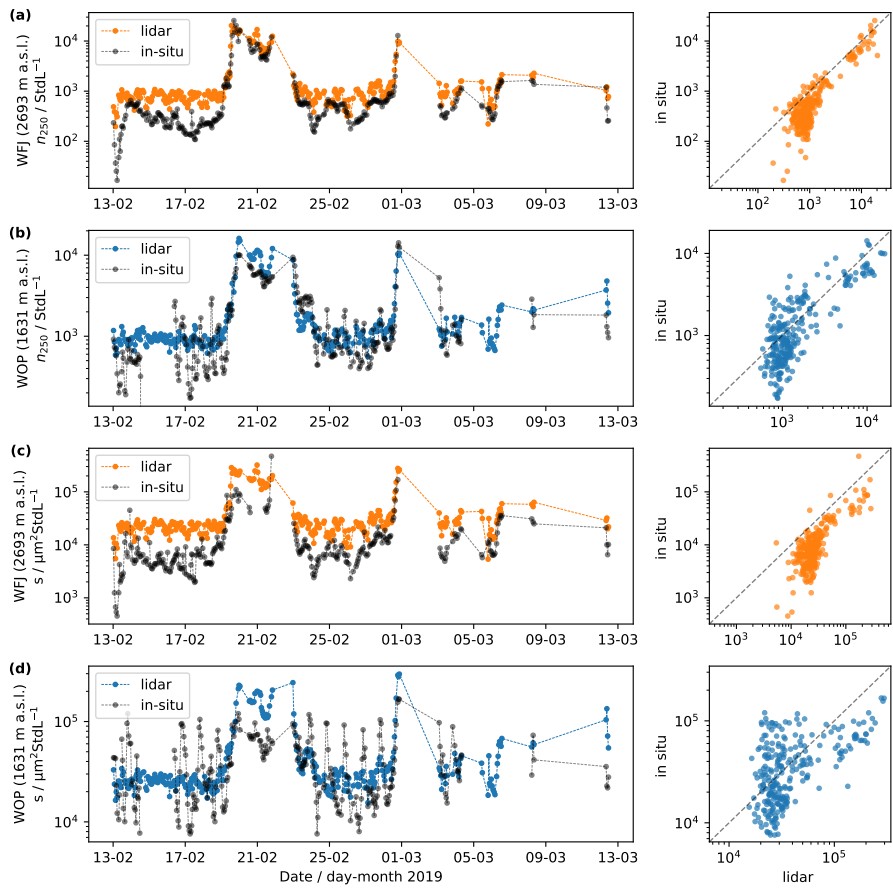

**Figure 2.** Comparison of aerosol properties observed in situ and with lidar. (a) Time series of aerosol number concentration with a radius larger than 250 nm ($n_{250}$) measured in situ (black) and with lidar (orange) at WFJ. (b) Same as (a) but measured at WOP (lidar data in blue). (c) Time series of surface area concentration ($s$) measured in situ (black) and with lidar (orange) at WFJ. (d) Same as (c) but measured at WOP (lidar data in blue). Right of each time series panel a scatter plot compares the in situ to the lidar observations of the data presented on the left. *Note: The lidar retrieval height at WOP was 2052 m a.s.l. (see Figure 1).*

bins at 2695 m a.s.l. and 2052 m a.s.l., respectively (Figure 1b). In the case of WOP, the lowest bin with complete overlap of the lidar (Wandinger and Ansmann, 2002, see e.g.,) was taken for the comparison, which is still around 400 m above the in situ site. The incomplete overlap of the emitted and received beam is a general issue in the comparison of lidar measurements to ground-based in situ observations. The comparison to WFJ, more than 1000 m above the lidar site where complete overlap is surely reached is a great advantage of the present study.

Figures 2a and 2b show $n_{250}$ from in situ observations and lidar retrievals for WFJ and WOP, respectively. Note, that the accuracy will be discussed only qualitatively. A quantitative assessment and improvement of the lidar aerosol retrieval is beyond the scope of this publication. For both locations in situ and lidar observations agree qualitatively well, in particular





for higher concentrations. At low concentrations, a plateauing of the lidar concentrations was observed due to the necessity of
setting a reference value for particle-free air in the lidar data retrieval algorithm. The backscattered signal is composed of the
air's molecular background and the contributions of aerosol particles. Observations in nearly particle free air (typically found
higher up in the troposphere) are used to calibrate the lidar to the molecular background. The theoretical minimal detectable
concentrations of continental aerosol during our field measurements were calculated to be 0.005 cm$^{-3}$ and 0.14 μm$^2$cm$^{-3}$,
respectively, using the lidar ratio and extinction conversion factors presented above and a reference value (estimated minimal
backscattered signal) of 0.001 Mm$^{-1}$sr$^{-1}$. In addition, deviations from the assumed linear relationship of extinction to $n_{250}$
and $s$, respectively, (Mamouri and Ansmann, 2016) could be relatively large for very low values of the extinction coefficient
($< 10$ Mm$^{-1}$) which were observed in the clear atmosphere over the Alps. The larger diurnal variability between the in situ
observation and lidar retrieval at WOP compared to WFJ can be explained by the diurnal changes of aerosol concentration near
the ground (in situ observations) not affecting the air masses on the lidar retrieval height (height difference approx. 400 m, see
Figure 1b). This difference in height and therefore air mass commonly limits a quantitative conclusion between ground-based
in situ observations and remote sensing instruments as the well-mixed boundary layer could at times not extend up to the
lowest retrieval height. For the retrieval of $s$ (Figures 2c and 2d) the aforementioned observations hold equally true, which is
not surprising as the surface area relates to the square of particle radius. However, comparing the retrieval accuracy at WFJ
(Figures 2a and 2c), a stronger bias of the lidar retrieved surface area concentrations is apparent which has been also previously
reported by Haarig et al. (2019), based on similar observations done at Barbados.

Despite the comparison in this section was purely qualitative, two important insights are gained: (i) Due to the better agree-
ment of in situ observations and lidar retrieval for $n_{250}$ compared to $s$, INP parameterizations basing on $n_{250}$ should be
preferred. (ii) Comparing ground-based in situ observations (collocated with the lidar instrument) do not allow for a compar-
ison even to the lowest lidar retrieval height. Consequently, we restrict the comparison to in situ INP concentrations in the
following to observations made at WFJ, where the lidar retrieval overlaps the in situ observations.

## 3.2 INP concentration comparison

Throughout the campaign, the region around Davos was mostly exposed to continental aerosol (e.g., 14–18 February 2019 as
seen in Figure 3). The situation changed 19–22 February 2019, when the synoptic wind situation promoted transport of Saharan
dust from North Africa towards Davos. The arrival of the dust plume is clearly visible in the attenuated backscatter coefficient
($\beta^{\mathrm{att.}}_{1064\mathrm{nm}}$) of the lidar (Figure 3a and 3b). Additionally, an increase by more than an order of magnitude in $n_{250}$ occurred at
both sites on 19 February (in situ observations, Figure 3c). Before 19 February, the higher variability of $n_{250}$ at WOP compared
to WFJ can be explained by local sources and accumulation of aerosol beneath a nighttime inversion along with valley and
mountain breezes (Wieder et al., 2021). Over the course of 19 February, $n_{250}$ at WOP followed the same trend and magnitude
as at WFJ and continued doing so in the following days. This drastic change in trend underlines that the region was affected
and dominated by the long-range transported Saharan dust. INP concentrations at $-13$ °C at WFJ also showed an increase of
approximately one order of magnitude already on 18 February (a day before the strong Saharan dust signal in Figure 3a and 3b).
INP concentrations could have already been higher on 17 February, however, no in situ measurements are available on that day.



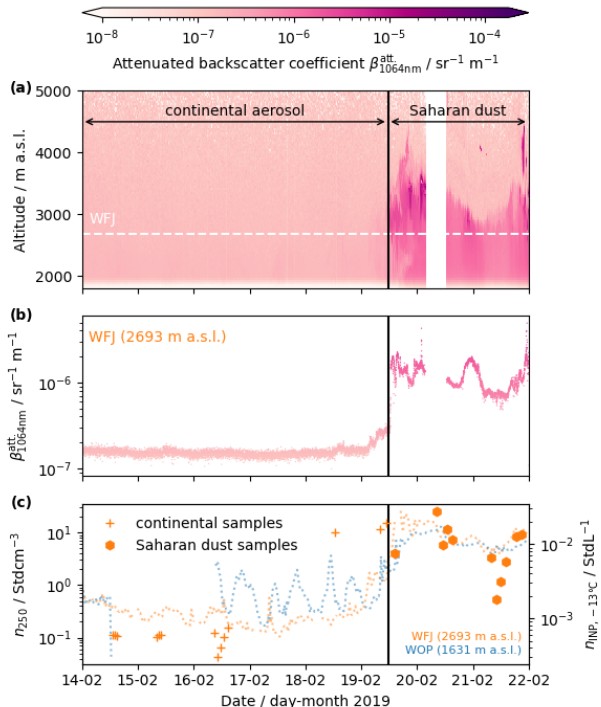

**Figure 3.** Overview of the Saharan dust event over the Davos region starting on 19 February 2019 and the continental aerosol conditions before. (a) Lidar attenuated backscatter coefficient ($\beta_{1064nm}^{att.}$) measured above WOP. Note that the gap in data on 20 February is due to technical maintenance of the lidar system. Retrieval height of WFJ is indicated in dashed white. (b) Lidar attenuated backscatter coefficient ($\beta_{1064nm}^{att.}$) retrieved from the lidar height bin closest to WFJ (2695 m a.s.l., Figure 1b). (c) In situ aerosol number concentrations for particles with radius larger 250 nm ($n_{250}$, left axis) at WFJ (dashed orange) and WOP (dashed blue) and INP concentrations at $-13\,°C$ ($n_{INP,-13°C}$, right axis) measured in situ at WFJ. INP samples during continental background are indicated with crosses and samples during Saharan dust are indicated with hexagons.

It is conceivable that air masses carrying only a smaller fraction of Saharan dust had already influenced the region of Davos. Analysis of the air mass properties and history prompted us to distinguish continental aerosol and Saharan dust for 19 February 11:30 UTC. The FLEXPART dispersion model (Pisso et al., 2019) indicated that air masses before that time originated from

Eastern Europe and Italy (not shown). Between 11 UTC and 12 UTC (time corresponding to measurements at Davos) the flow over Southern Italy was affected by intrusion of air masses from North Africa. To identify when exactly air masses from North Africa carrying Saharan dust entered the flow we utilized the relative humidity measured at WFJ (Figure A1a). A local minimum in relative humidity was identified on 19 February at 11:30 UTC. Therefore, we classify samples before that time still as continental. Nonetheless, the Jungfraujoch Research Station (approximately 140 km WSW of Weissfluhjoch) released

a Saharan dust event warning only at 13:37 UTC. Brunner et al. (2021) reported that the determination of a Saharan dust event varies based on the underlying parameter used as proxy. The three samples with higher INP concentrations (orange crosses





**Table 2.** Overview of the number of samples ($N$) available for dust-dominated and continental air mass cases. Additionally, the number of days over which the samples are taken is given.

| Air mass | $N$ | days |
|---|---|---|
| dust | 9 (11) | 3 |
| continental | 23 | 14 |

on 18 and 19 February in Figure 3c) coincided with local maxima in relative humidity and attenuated backscatter (indicated by 1 and 2 in Figures A1a and A1b, respectively). Therefore, the higher INP concentrations observed at WFJ at the end of the continental aerosol period could be a result of a stronger exchange with air masses at lower height resulting in uptake of

biogenic and soil particles from fertile lands over Southern Italy and Eastern Europe (Conen et al., 2015).

In the following we will investigate the performance of INP parameterizations for dust and continental aerosol air masses. For the dust comparison (Section 3.2.1) we use the nine of the 11 in situ samples indicated by hexagons in Figure 3c. For the two samples in the morning of 20 February, no lidar data is available for a comparison. For the comparison with continental aerosol (Section 3.2.2) we use the 14 samples presented with crosses in Figure 3c and additional samples that were collected

individually during clear sky conditions before and after cloud events between 23 February and 12 March 2019, resulting in a total of 23 samples collected on 14 days (see Table 2). The grouping into Saharan dust and continental samples is consistent with Mignani et al. (2021) (cf. supplementary material Figure S1).

In the comparison between in situ and lidar observations, four sources of uncertainty are present: (i) the measurement uncertainty of the in situ observation (drop freezing technique) which depends on various parameters such as freezing temperature

and number of droplets. Vali (1971) states the uncertainty to be between a factor two and four (based on an assay of 150 droplets). (ii) the retrieval uncertainty of the lidar that are passed on to the INP concentration through the applied parameterization. The retrieval uncertainty of the INP concentration from the lidar measurements is given by a factor 3–10 (Mamouri and Ansmann, 2016). (iii) Air mass differences due to the horizontal distance between in situ measurements at WFJ and the lidar beam which was 3.65 km (Figure 1b). However, being located on mountaintop height, it is conceivable to assume no drastic

difference between the two locations. Thus, the uncertainties caused by the spatial distance are negligible compared to the other sources of uncertainties. Ultimately, (iv) the natural variability of INP is typically referred to as one order of magnitude (Kanji et al., 2017). The latter being the largest source of uncertainty, we consequently view data points of in situ measurements and lidar based observations within one order of magnitude as acceptable.

### 3.2.1 Retrieval during times of Saharan dust

In situ INP observations are compared against lidar-retrieved INP concentrations using different INP parameterizations to evaluate their performance during Saharan dust presence. In Figure 4, we present comparisons to the parameterizations of D15, H19, and U17d along with performance measures such as the mean absolute error (MAE), the fraction of data points





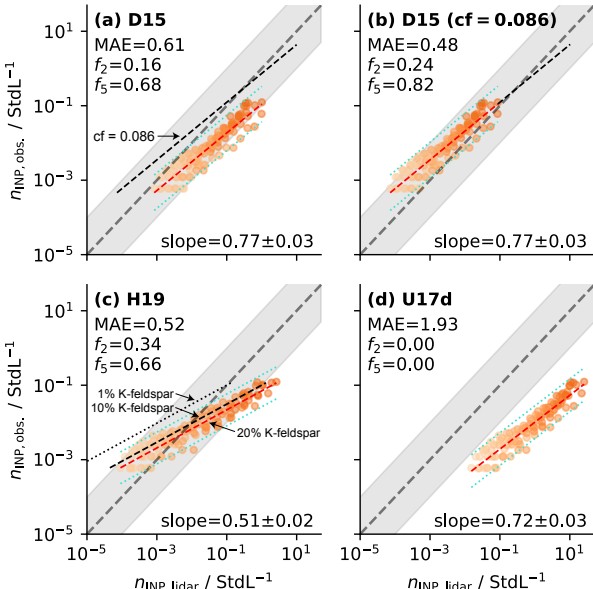

**Figure 4.** Comparison of in situ INP concentrations ($n_{\mathrm{INP,obs.}}$) measured at WFJ to lidar-retrieved INP concentrations ($n_{\mathrm{INP,lidar}}$) during the Saharan dust event (19–21 February 2019) for three dust parameterizations: (a) D15 (DeMott et al., 2015), (b) D15 with a calibration factor (cf = 0.086) applied as proposed by Schrod et al. (2017), (c) H19 (Harrison et al., 2019), and (d) U17d (Ullrich et al., 2017). Notably, we compare INP concentrations at temperatures between $-20\,°\mathrm{C}$ (dark orange) and $-5\,°\mathrm{C}$ (light orange). The 1:1 line is shown in dashed gray. The dashed red lines show a linear regression fit through the logarithmically transformed data points, including the 95% confidence interval in dotted cyan. Shown in every panel are the mean absolute error (MAE), the fraction of data points falling within a factor of two and five with respect to the 1:1 line ($f_2$ and $f_5$), and the slope of the fit (including the standard error). The gray shaded area indicates a deviation of one order of magnitude from the 1:1 line which is the upper limit uncertainty of the lidar based INP retrieval (Mamouri and Ansmann, 2016). For D15 (a), the data fit is also presented applying the calibration factor from (b) for comparison (black dashed). For H19 (c), the data fit is also presented assuming a K-feldspar fraction of 10 % (black dashed) and 1 % (black dotted), respectively.

falling within a factor of two and five with respect to the 1:1 line ($f_2$ and $f_5$), and the slope of a linear regression performed on the logarithmically transformed data. Among all investigated parameterizations, D15 (Figure 4a) results in a slope of the linear

regression fit that is closest to the desired unity. However, D15 overestimates the INP concentration, which could partially be attributed to the slight overestimation in $n_{250}$ retrieval (Figure 2a). Schrod et al. (2017) proposed a calibration factor of cf = 0.086 based on a lidar comparison with Saharan dust samples collected with UAVs over Cyprus. Applying this calibration factor to the data pushes the data beyond the 1:1 line, reducing the MAE and increasing $f_2$ and $f_5$ (Figure 4b). Previously, Mignani et al. (2021) compared INP observations at $-15\,°\mathrm{C}$ of the same samples to predictions of D15 using in situ aerosol

data and applying the calibration factor of Schrod et al. (2017). In this study, we extend the analysis to the whole temperature spectrum ($-20\,°\mathrm{C}$ to $-5\,°\mathrm{C}$) and relied on aerosol retrievals from the lidar. For concentrations between $10^{-1}$ and $10^1\,\mathrm{StdL}^{-1}$, which is the concentration range Schrod et al. (2017) used to determine the calibration factor, the fit cuts the 1:1 line (black





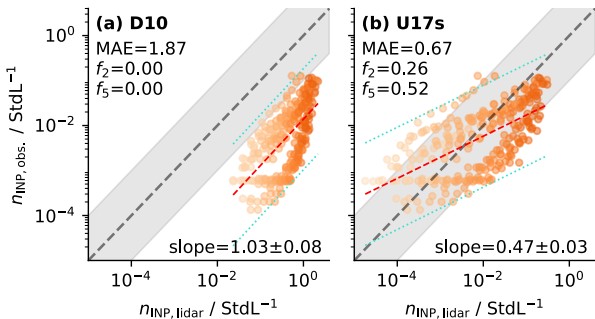

**Figure 5.** Same as for Figure 4 but during continental background (cf. Figure 3) for the two parameterizations: (a) D10 (DeMott et al., 2010), and (b) U17s (Ullrich et al., 2017).

dashed extension of the red dashed data fit). Note that Schrod et al. (2017) also proposed a slope correction, however, we did not correct the slope because all data points already fall within the uncertainty range (gray shaded area). H19 (Figure 4c) results

in a higher $f_2$ as D15 (using $\mathrm{cf} = 0.086$) but with a more shallow slope leading to a lower $f_5$. For the retrieval we assumed a K-feldspar fraction of 20% in accordance with Kandler et al. (2011) who investigated the relative K-feldspar fractions on Cape Verde west of Central Africa. H19 scales log-linearly with the percentage of K-feldspar. Kandler et al. (2018) found during measurements at Barbados that Saharan dust featured only 1% K-feldspar. However, using 1% K-feldspar results in an underestimation of INP concentration (Figure 4c). A fraction of approximately 10% K-feldspar would match our observations

best, yet the slope of the relation would be too small (0.51). U17d (Figure 4d) overestimates the INP concentration the strongest leading to the highest MAE (1.93) among the investigated parameterizations.

     Our comparison suggests that D15 in combination with the calibration factor ($\mathrm{cf} = 0.086$) proposed by Schrod et al. (2017) as the preferable choice for INP concentration retrieval from dust air masses (slope closest to unity, lowest MAE). Moreover, D15 should be preferred due to its dependence on $n_{250}$ instead of $s$ which is associated with lower lidar retrieval uncertainty

(see Section 3.1).

### 3.2.2    Retrieval during times of background aerosol

Aside from a Saharan dust event, the measurement region was not affected by a long-range transported dominating aerosol species. Thus, the samples collected outside the Saharan dust episode represent a general mix of continental aerosols. In Figure 5, we compare the in situ observed INP concentration of these samples with lidar retrievals using the parameterizations: (a)

D10 for global aerosol and (b) U17s for soot which are the best parameterizations available to mimic continental aerosol. D10 results in a large MAE (1.87) and with no data points agreeing within a factor of two or five ($f_2 = f_5 = 0$). Yet, the slope is close to unity (1.03). INP concentrations based on U17s result in smaller MAE (0.67) and higher $f_2$ (0.26) and $f_5$ (0.52), but the slope is more shallow (0.47). Purely based on the error assessment of the parameterizations in their original form, U17s would be the parameterization of choice for continental INP concentration lidar retrieval. The slope of the D10 comparison





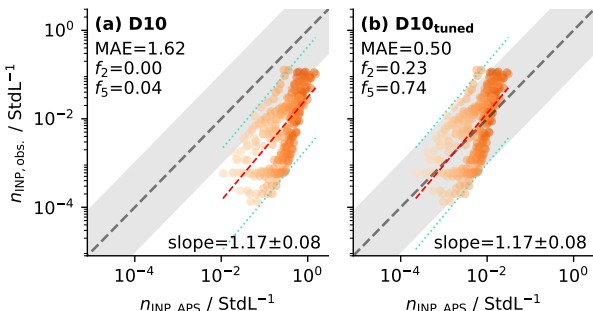

**Figure 6.** Same as for Figure 5a but comparing in situ INP concentrations ($n_{\text{INP,obs.}}$) measured at WFJ to predicted INP concentrations ($n_{\text{INP,APS}}$) based on D10 now applied to in situ aerosol concentrations (from APS) during continental background (cf. Figure 3). (a) D10 applied to in situ aerosol concentrations (from APS) in its standard form. (b) D10 shifted towards the 1:1 line maximizing the fraction of data points within one order of magnitude around the 1:1 line (cf. Equation 1).

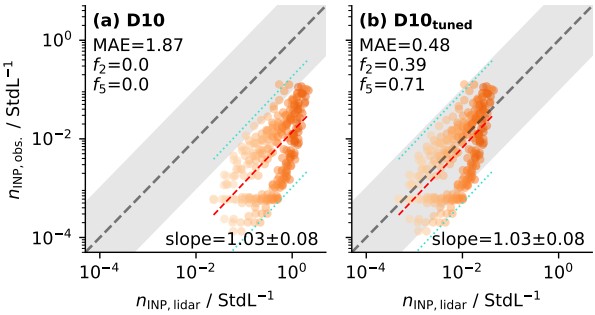

**Figure 7.** Same as for Figure 4 but comparing in situ INP concentrations ($n_{\text{INP,obs.}}$) measured at WFJ to lidar-retrieved INP concentrations ($n_{\text{INP,lidar}}$) using (a) the standard D10 parameterization and (b) the tuned D10 parameterization (see Equation 1) for continental aerosol.

being close to unity indicates that D10 captures the composition of a continental aerosol mix. Together with the large MAE, the comparison suggests that the expected fraction of INPs among the aerosol mix is overestimated. In the next section, we propose a tuning of D10 to improve the retrieval of continental INP concentration from lidar measurements.

### 3.2.3   Tuning of D10 based on in situ INP and aerosol measurements

The slope in the comparison between in situ observations and lidar retrieved INP concentrations suggests that D10 captures the
feature of a continental INP mix (Fig. 5a). The large overprediction by nearly two orders of magnitude on average (MAE=1.87) indicates an overestimation in INP active aerosol fraction. D10 was developed based on nine data sets from different locations (e.g., Alaska, Pacific, Amazon Basin) and thus contains various aerosol types (DeMott et al., 2010). Although a large number of non-dust samples contributed to the parameterization, dust samples e.g., from the Pacific Dust Experiment (PACDEX, Stith





et al., 2009) were included in the D10 parameterization. Mineral dust is more ice active compared to (non-biogenic) continental

aerosol. Therefore, it is not surprising that D10 overestimates the INP concentration when used for non-dust continental aerosol. Using the collocated in situ observations at WFJ of $n_{250}$ (from APS) and INP concentration (Section 2.1), we could derive a tuned D10 parameterization optimized for continental aerosol. The optimization is based purely on the in situ observations in order to be independent of the lidar data (which is the test data) as we cannot fully rule out a difference in air mass. Analogously to the lidar retrieval, INP concentration is predicted by the aerosol particle number concentration of particles with radius larger

250 nm (now taken from the APS) and D10 applied to it. The comparison of the obtained INP concentrations (Figure 6a) exhibits very similar performance measures as in the lidar comparison (Figure 5a) underlining that the difference in air mass was minuscule. We optimize D10 with a multiplicative calibration factor (cf) as a free parameter (similarly to DeMott et al., 2015; Schrod et al., 2017) for maximizing the number of data points within an order of magnitude around the 1:1 line (gray shaded area in Figure 6b). Based on the performance parameters it is apparent how the prediction has improved while the slope

remained unchanged. To obtain the tuned INP concentrations ($n_{\mathrm{INP,D10,tuned}}$) from INP concentrations using D10 ($n_{\mathrm{INP,D10}}$), we calculated

$$n_{\mathrm{INP,D10,tuned}} = \mathrm{cf} \cdot n_{\mathrm{INP,D10}} \tag{1}$$

with the calibration factor, $\mathrm{cf} = 0.0204$, and the INP concentrations given in $\mathrm{StdL^{-1}}$. The calibration factor indicates that under clean continental aerosol conditions just 2 % of the INP predicted with D10 are present. The improvement in prediction

that was found using the in situ aerosol data is also found for the lidar retrieval when applying the calibration factor (Figure 7). The now tuned D10 parameterization outperforms U17s (compare Figure 5b and 7b). Thus, we propose the tuned D10 parameterization (Equation 1) to retrieve atmospheric INP concentrations from wintertime continental air masses.

### 3.3 Validation of remotely retrieved cloud properties

In this section, remotely retrieved IWCs and ICNCs are compared with balloon-borne in situ observations to validate the

accuracy of the retrievals. On 8 March 2019, the balloon performed vertical profiles between 1650 and 1900 m a.s.l. next to the remote sensing instruments (see overview plot in Ramelli et al., 2021a, Figure 6b). Since the lowest data bin of the radar and lidar was at 1783 m a.s.l. we considered in situ data only for times when the balloon was at an elevation of 1750 m a.s.l. and higher (gray shading in time series plots of Figure 8, maximum elevation 1910 m a.s.l.). IWC is deduced solely from radar observations and could continuously be taken from the lowest height bin (1783 m a.s.l.). The retrieval of ICNCs needs parallel

measurements of the radar and lidar. As the lidar featured data gaps, it was necessary to average (median) the ICNC data between 1900 and 2500 m a.s.l.. The uncertainty of remotely retrieved cloud properties is typically given by a factor of three (Bühl et al., 2019). Within a factor of three, 68 % and 46 % of the data points agreed for IWC (Figure 8a) and ICNC (Figure 8b), respectively. For a factor of ten, the agreement was 92 % and 88 % for IWC (Figure 8a) and ICNC (Figure 8b), respectively. Given the inhomogeneous distribution of hydrometeors within a MPC (Korolev et al., 2017) and that the balloon could have

been up to 200 m horizontally displaced from the remote sensing beams, we assess the comparability to be satisfactory. In the following analysis we will consider remotely retrieved ICNC associated with an uncertainty of one order of magnitude.





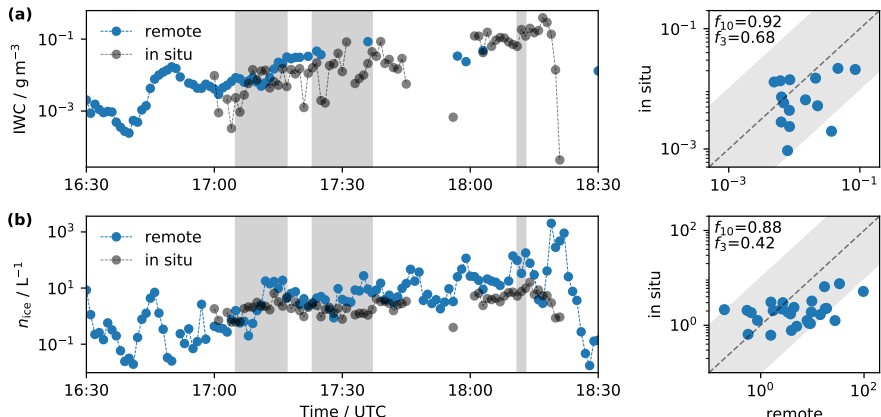

**Figure 8.** Comparison of the remotely retrieved and in situ measured (a) ice water content (IWC) and (b) ice crystal number concentration ($n_{\text{ice}}$) on 8 March 2019. The IWC data was retrieved from the lowest size bin (1783 m a.s.l.). Due to measurement gaps, ICNC between 1900 and 2500 m a.s.l. was averaged (median). Left: Time series of available remotely retrieved data (blue) and in situ observed data (black). Right: Scatter plot of data during times the balloon was at an elevation higher than 1750 m a.s.l. (gray shading in the time series plots, maximal elevation: 1910 m a.s.l.). The gray shading in the scatter plots indicates an error of one order of magnitude. The values $f_{10}$ and $f_3$ indicate the fraction of data points within a factor ten and three, respectively.

## 3.4 IMF estimation in orographic MPCs

Here we propose a method to assess SIP on a single cloud basis by calculating IMF histograms using combined lidar and radar retrievals. First, the procedure is explained for one observed cloud case (Section 3.4.1). Second, the effect of sublimation
below cloud base on ICNC and the temporal evolution of ICNC profiles is presented. Observed IMF histograms with height and temperature are presented and discussed with regard to the ICNC input data (Section 3.4.2). Third, the limitations and caveats of the method are discussed (Section 3.4.3). Finally, the results of this analysis are compared to observations of previous field measurements (Section 3.4.4).

### 3.4.1 Procedure description

Locally in a cloud, the magnitude of ice crystal enhancement can be estimated with the IMF defined as ICNC divided by INP concentration. Note that the effect of blowing snow is neglected and sedimenting ice crystals from a cloud layer aloft are ruled out based on manual inspection of cloud radar observations. For this analysis, only ICNCs in ice saturated regions were considered, as otherwise ICNCs can decrease due to sublimation. The INP concentration is indicative of the amount of primary ice crystals formed in the cloud that could evoke subsequently SIP. Taking a too low INP concentration results in an
overestimation of the IMF. The (cumulative) INP concentration increases with decreasing temperature, resulting in the highest INP concentrations at cloud top (assuming that the entire cloud formed by the same ascending air mass). ICNCs at cloud top



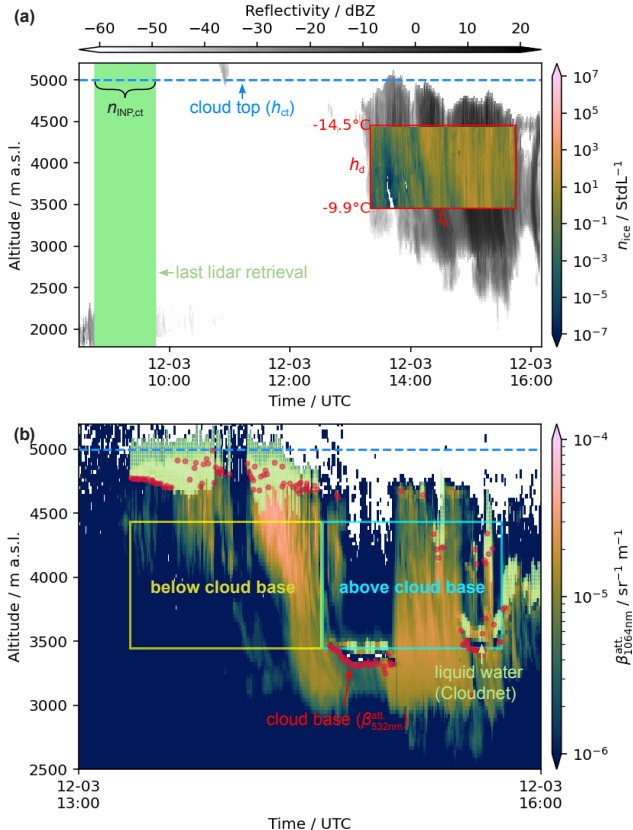

**Figure 9.** Exemplary illustration of data processing workflow to obtain in cloud IMFs for the cloud observed on 12 March 2019. (a) Reflectivity of the cloud radar is given in shades of gray. ICNC ($n_{ice}$) is overlayed in color. The limit of used data for the IMF calculation with height is indicated in red (height span $h_d$, time span $t_d$). Additionally, the ambient temperature at the heights of the upper and lower bound of selected data is given in red (see $T_{dt}$ and $T_{db}$ in Table D1). The INP concentration ($n_{INP,ct}$) was taken from the last lidar retrieval (solid green shading) at cloud top (dashed blue line). (b) Attenuated backscatter coefficient at 1064 nm ($\beta_{1064nm}^{att.}$) from the lidar in color. Cloud base estimates determined from the attenuated backscatter coefficient at 532 nm ($\beta_{532nm}^{att.}$) are indicated by red dots. Regions where the Cloudnet algorithm detected liquid water are indicated in bright green. The height of cloud top is indicated by the dashed blue line. The region within the yellow rectangle was classified as below cloud base and the region within the cyan rectangle as above cloud base, respectively. Overviews of all individual cloud events can be found in Figures B1 and C1.

are generally created due to primary ice formation (see e.g., Crosier et al., 2014). At sufficient size, the primary ice crystals will no longer be levitated by the updrafts and start to sediment to lower heights. Thus, using INP concentrations at cloud top is an upper estimate for the primary ice crystals in the cloud and avoids an overestimation of the IMF. In the next section, we

determine IMFs for seven individual cloud events based on the following procedure:

1. Based on the aforementioned considerations, we retrieve the INP concentration at cloud top height from the lidar retrieval closest to the cloud event (Figure 9a), using the tuned D10 parameterization (Equation 1), which is the best estimate using





ground-based lidar. ICNCs were taken over a height $h_d$ (Figure 9a), excluding heights near the radar echo boundaries to avoid retrieval errors and to avoid zones of cloud top entrainment. Additionally, data was trimmed to a time $t_d$ (Figure 9a) to account for echo free regions at the beginning and end of the clouds. Echo free regions in the selected data frame (e.g., on a height of approximately 3600 m a.s.l. at around 13:40 in Figure 9a) are thought to be negligible for the consequent statistical analysis.

2. Ice crystals sedimenting out of the cloud will start to sublimate upon entering the unsaturated environment below cloud base. Thus, in a second step, lidar variables are used to determine the cloud base height to discard data from regions where sublimation could occur. In our study, the classification was done manually by inspecting (i) the attenuated backscatter coefficient at 1064 nm ($\beta^{\mathrm{att.}}_{1064\mathrm{nm}}$), (ii) a cloud base estimate derived from the attenuated backscatter coefficient at 532 nm ($\beta^{\mathrm{att.}}_{532\mathrm{nm}}$), and (iii) the presence of liquid water flag by the Cloudnet algorithm (Figure 9b). In most cases criteria (ii) and (iii) were used to determine cloud base because these estimates are deemed to be more reliable than criteria (i). However, in cases where these estimates provided frequent jumps in cloud base, for example, due to snow fall blurring the backscattered signal (see e.g., 15:00 UTC - 15:30 UTC in Figure 9b), we used (i) as a proxy for the cloud base. As the cloud base height in general changes with cloud evolution, the data was divided into time slices of approximately one hour for which individually averaged cloud base heights were determined. The individual sizes of the time slices were set to match distinctive regime changes (e.g., the lowering of the cloud base in Figure 9b). Consequently, each time slice was vertically divided and classified into *below cloud base* and *above cloud base*. Strong precipitation can hamper the lidar signal close to the instrument, disallowing cloud base determination above. Time slices during strong precipitation or without lidar information are therefore classified as *unknown*.

3. The IMF at a given height $h$ and time $t$ is consequently calculated as

$$\mathrm{IMF}(h,t) = \frac{n_{\mathrm{ice}}(h,t)}{n_{\mathrm{INP,ct}}} \tag{2}$$

with $n_{\mathrm{ice},h,t}$ being the ICNC at height $h$ and time $t$ and $n_{\mathrm{INP,ct}}$ being the INP concentration at cloud top (as illustrated in Figure 9a).

### 3.4.2 IMF observations of seven orographic MPCs

IMFs were calculated for seven cloud events according to the procedure described above. An overview of the ICNC data and the structure of each cloud is presented in Appendix B. The time slice division and classification are provided in Appendix C. Our methodology uses a fixed INP concentration for all IMFs (Equation 2). Thus, changes in the IMFs are driven by the evolution of the ICNCs. In a generalized case of cloud evolution, one would expect the ICNCs to increase with time due to SIP as long as conditions for SIP prevail. At cloud top primary ice crystals will nucleate, start growing, and eventually sediment. Due to the sedimentation, ICNC increases downward. If the environmental conditions permit, SIP processes start to enhance the ICNC. With the presence of more ice crystals and warmer temperatures, aggregation becomes more likely, reducing the ICNC towards cloud base and at higher temperatures. With ice crystals sedimenting below cloud base, not only aggregation



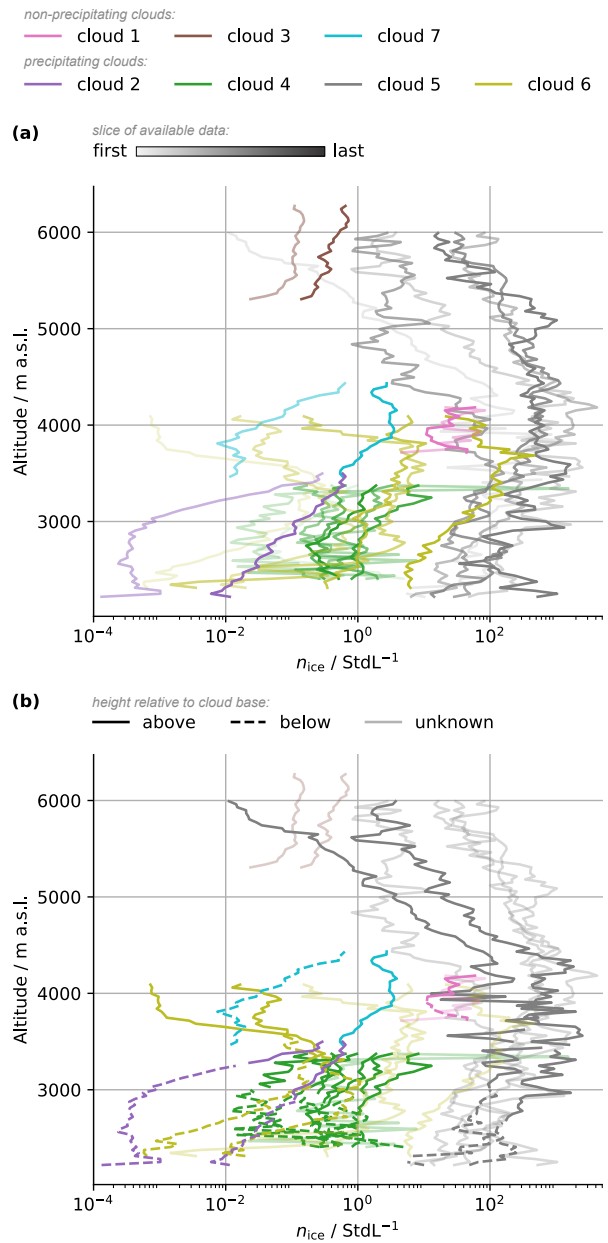

**Figure 10.** Median ICNCs as a function of altitude for each cloud and for time slices of approximately one hour. (a) Temporal evolution of the ICNC profiles: Faintest colors correspond to the first slice of data and the strongest colors correspond to the last slice of data. (b) Cloud regime classification for each time slice: ICNCs taken above cloud base (solid), below cloud base (dashed) and unknown cloud base (faint color). For cloud 3, the cloud base could not be determined.





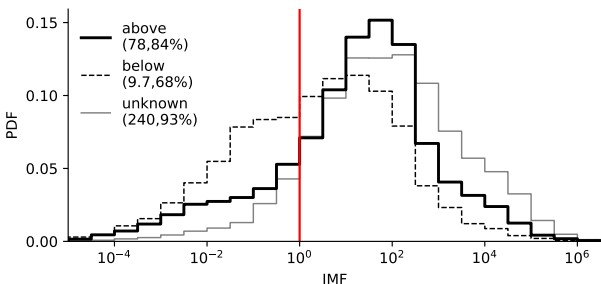

**Figure 11.** Probability density functions (PDFs) of obtained IMFs (Equation 2) for ICNC data above cloud base (solid black), below cloud base (dashed black), and of unknown cloud base (solid gray). The solid red line indicates unity. The median IMF and the percentage of IMF larger unity for each PDF are given in the legend.

but also sublimation of entire ice crystals could reduce the ICNC. In this simple picture, the recirculation of ice crystals back into the cloud due to updrafts is not considered. In reality it is not possible a priori to know where in the cloud ICNCs increase or decrease as this strongly depends on the environmental conditions and hence varies from cloud to cloud and throughout a cloud's lifetime.

Figure 10 gives an overview of the median ICNC profiles for each of the seven observed clouds in this study as time slices of approximately one hour. A general trend of increasing ICNC with time manifests itself across different clouds (e.g., clouds 2, 3, 5, 6, and 7). For clouds 5 and 6 ICNCs increase from cloud top downwards before they start to decrease towards the ground. In contrast, the ICNC profiles of the other clouds (clouds 1, 2, 3, 4, and 7) generally decrease with decreasing height. Note that the top of the profile is below cloud top (see discussion in Section 3.4.1). Above cloud base, aggregation can reduce the ICNC. Below cloud base, sublimation can reduce the ICNC in addition to aggregation (Figure 10b). Despite no cloud base determination, ICNC profiles of clouds 5 and 6 during strong precipitation increase with time near the surface (especially for cloud 6), suggesting that the relative humidity was close to 100 % with respect to ice everywhere below cloud. Note that for cloud 5 in the last hour, there could be ice crystals originating from homogeneous freezing sedimenting from higher up (see Figure B1d). Cloud base for cloud 3 could not be determined due to a liquid layer in a lower cloud fully attenuating the lidar signal (see Figure A1b). It is worth mentioning that the ICNCs do not differ strongly between precipitating and non-precipitating clouds.

Following the definition of the IMF (Equation 2), those below unity are linked to regions in the cloud where the INP concentration at cloud top exceeds the local ICNC. Potential regions of IMF close to and lower than unity are (i) in the beginning stage of cloud formation and (ii) near cloud top, where primary ice crystals have just formed and SIP processes are not active yet. Furthermore, ICNC and thus IMF could be reduced (iii) towards cloud base due to aggregation and (iv) below cloud base due to sublimation of ice crystals. Note that sublimation could also act as a SIP process as ice crystals could fragment during sublimation (Korolev et al., 2020). The obtained IMF distributions for the three categories (above cloud base, below cloud base, and unknown cloud base) for all clouds combined are presented in Figure 11. Above cloud base, IMFs > 1





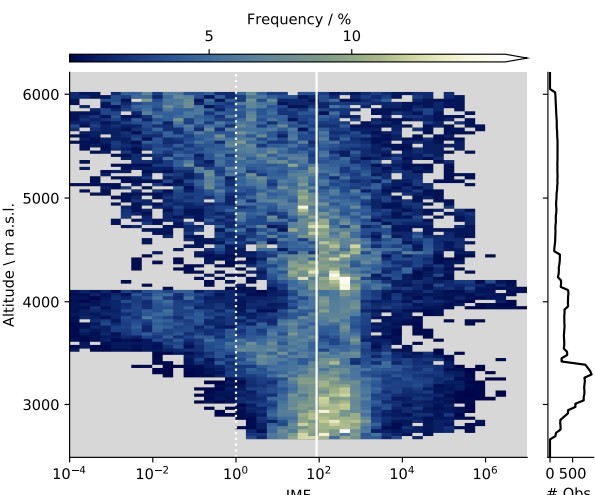

**Figure 12.** Frequency histogram of all observed IMF above cloud base combined with height. Frequencies are normalized to the number of observations (right axis) per height bin. Gray background indicates where no ICNC data was observed. The dashed white line indicates an IMF of unity. The median IMF of all data is indicated in solid white.

(and thus active SIP) were observed nearly 84% of the time with a median IMF of around 80 and an interquartile range of two orders of magnitude (see Table D1). As expected, IMFs below cloud base are generally found at lower values along with a wider

spread than above cloud base. In contrast, the distribution of IMF where no cloud base could be determined is shifted to even higher values than the IMF distribution above cloud base (median IMF higher by approximately a factor of three). Additionally, the shape resembles the above cloud base distribution. These observations suggest that ICNCs in the unknown category were potentially not strongly affected by sublimation and could be classified as above cloud base. Table D1 summarizes the IMF calculations per cloud as well as individual key numbers of the clouds (e.g., concentration of INP and aerosol at cloud top, and

temperatures at different heights). The individual IMF distributions per cloud and all calculated IMFs combined are presented in Figure D1. In Figure 11, it is conspicuous that the IMF distribution (above cloud base) features a heavy tail below unity signifying higher INP concentration at cloud top than local ICNCs. These IMFs can be explained when looking at the IMFs as a function of height (Figure 12). IMFs below unity are found in two height ranges (one between $3500\,\mathrm{m\,a.s.l.}$ and $4000\,\mathrm{m\,a.s.l.}$ and one between $4500\,\mathrm{m\,a.s.l.}$ and $6000\,\mathrm{m\,a.s.l.}$). The two regions belong to the early stages of clouds 5 and 6 (see Figure

10a) where the cloud was still developing. This suggests that regions of IMFs smaller than unity could be a proxy for regions where primary ice generation dominates. After the first hour, median ICNCs of clouds 5 and 6 increased by over two orders of magnitude and one order of magnitude, respectively. Accordingly, also the IMFs increased at the corresponding heights above unity, potentially resembling a change from a primary ice dominated regime towards the onset of SIP.

Despite all uncertainties related to SIP, the effectiveness of the underlying processes varies with temperature (e.g., the

Hallett-Mossop process is thought to be active only between $-8\,°\mathrm{C}$ and $-3\,°\mathrm{C}$). To elucidate the obtained IMF with regard to



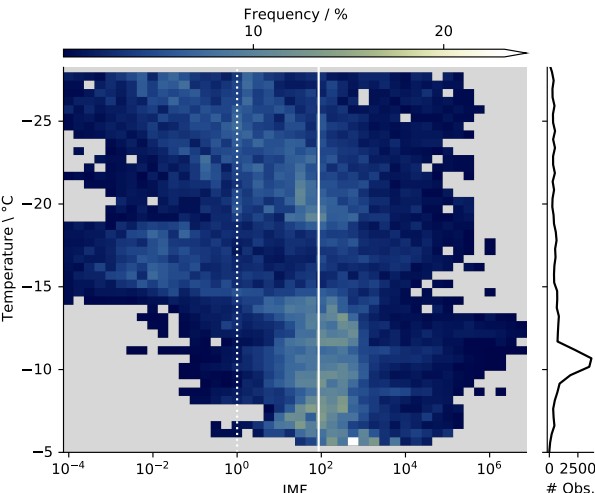

**Figure 13.** As for Figure 12, but with the height converted to temperature using data from the COSMO-1 analysis.

ambient temperatures, retrieval height was converted to ambient temperature using the COSMO-1 analysis. The temperatures at top and base of the data frame were averaged over the time of available ICNC data (as listed in Table D1 and indicated in Figure B1) and interpolated linearly with height. The resulting histogram of IMFs from all cloud events is presented in Figure 13. Individual histograms per cloud can be found in Figure E1. SIP was active in all observed clouds (IMFs > 1) and at all

observed temperatures highlighting that SIP processes are not only restricted to warm subfreezing temperatures, where the Hallett-Mossop process prevails. High frequencies of IMF above the median IMF were observed between $-10\,°C$ and $-5\,°C$. Hanna et al. (2008) found that for cloud tops at temperatures between $-10\,°C$ and $-10\,°C$ frequently precipitation is initiated. Strong SIP leading to high ICNC could promote precipitation initiation given a favorable environment. However, no substantial difference in IMF distributions between precipitating and non-precipitating clouds was found (highest IMFs were even found

for a non-precipitating cloud, cloud 7 in Figure E1). Such a link of the observed IMFs to the ambient environmental conditions is not made since our measurements are restricted to only seven cloud events, with the number of observations biased towards temperatures warmer than $-15\,°C$.

### 3.4.3   Uncertainties and caveats of the proposed methodology

As stated in the introduction, using the IMF itself does not allow to infer the underlying SIP process due to the potential

(temporal and spatial) displacement of first ice crystal formation and occurrence of SIP. For the same reason, it is also not possible to quantitatively describe the uncertainty of the calculated IMFs. Three main factors add uncertainty to this method: (i) Uncertainties from the ICNC retrieval which were shown to be within an order of magnitude (Figure 8). (ii) Uncertainties from the INP concentration retrieval at cloud top which are also within an order of magnitude (Figure 7b). (iii) Uncertainties from assuming a constant INP concentration with time throughout the cloud. Based on the retrieved INP concentrations for the





seven clouds (Table D1) ranging from approximately $10^{-3}$ and $10^{-1}$ $\mathrm{StdL^{-1}}$ we estimate this uncertainty to also be of one order of magnitude. The combined influence of the uncertainties on the calculated IMF is not possible to determine due to the missing spatial (and thus logical) link between cloud top INP concentration and in cloud ICNC following the considerations above.

Our method has additional caveats:

– All ICNC data above cloud base is assumed to not be affected by sublimation. However, it is not necessarily true that above the detected cloud base no sublimation occurred as dry air layers could have been entrained.

– In most SIP processes, small ice splinters are created that could then start growing in favorable cloud conditions. This small ice would create a small but sharp peak at small velocities in the Doppler velocity spectrum (see e.g., Figure 1c in Li et al., 2021). In the ICNC retrieval, the entire Doppler velocity spectrum is fit to an unimodal distribution for an assumed ice crystal shape (Bühl et al., 2019). Consequently, the secondary ice would only considerably affect the retrieved ICNC when the ice splinters have grown to a size when their distribution approaches and ultimately merges with the background ice distribution.

– Excluding ICNC found in sublimation conditions could exclude contributions of the SIP process of sublimation fragmentation (Korolev et al., 2020). However, the produced secondary ice crystals will not fully sublimate if they reenter the cloud again where they would contribute to the ICNC above cloud base.

– In our approach, a constant cloud top height and thus a constant INP concentration at cloud top was assumed for an entire cloud event. We tested that the effect of individual cloud heights did not substantially affect the obtained IMFs.

– Calculating the ICNC from the remote sensing observations requires a sufficiently strong detectable lidar echo limiting our method to optically thinner clouds (where the signal is not attenuated) which could induce a bias. Thus, all quantitative findings above need to be treated with caution. The statement that can be conveyed with higher certainty is the observation of SIP being active at all temperatures.

The data of each cloud was selected manually (see red data frames in Figure B1). A robustness test of the data selection is provided in Appendix F.

### 3.4.4 Comparison to previous field observations

Atmospheric ICNC and INP concentrations have been studied for more than 50 years, most often utilizing aircraft as a measurement platform. In the beginning of the millennium it was brought to the cloud physics community's attention that previous measurements could have recorded artificially enhanced ice crystal concentration due to shattering of ice crystals on the tips of the cloud probes (see e.g., Field et al., 2006; Korolev et al., 2011). As a response, correction algorithms were developed (e.g., Field et al., 2006; Lawson, 2011) and the probe design was adapted (e.g., Korolev et al., 2011, 2013). Thus, we compare
our findings only to field observations of selected studies from the past decade in which the effect of shattering was mitigated.





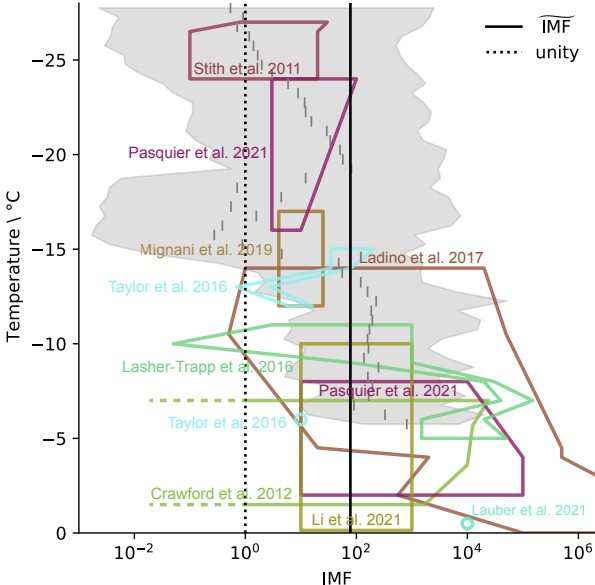

**Figure 14.** Comparison of obtained IMFs in this study (gray shading indicating the $10^{th}$ and $90^{th}$ percentile of observations with the median indicated by vertical lines per temperature bin) to previous field observations. The polygons envelop the observed IMFs per study. Circles indicate observations of one IMF at one temperature. Note that for the study of Crawford et al. (2012), no lower limit in IMF could be determined. Note that the retrieved IMFs from Ladino et al. (2017) tend towards $10^9$ for temperatures towards 0 °C. The dashed black line and the solid black line indicate an IMF of unity and the observed median IMF in this study, respectively.

Details about the individual studies and the procedure of obtaining IMFs from the ice crystal and INP data are described in Appendix G.

Figure 14 comprises the range of observed IMFs in this study along with derived IMFs from previous studies which observed maritime convective systems (Stith et al., 2011), continental clouds (Crawford et al., 2012; Taylor et al., 2016), tropical clouds

(Lasher-Trapp et al., 2016; Ladino et al., 2017), orographic MPCs (Mignani et al., 2019; Lauber et al., 2021), and Arctic MPCs (Pasquier et al., 2021). The IMFs derived from the previous studies exhibit not only great variability among each other but also span over a considerable value range in each individual study. The ranges of IMF from previous studies generally coincide with the observations from our study, strengthening the applicability and validity of our method. Most studies observed clouds at temperatures warmer than −15 °C and some acknowledge the Hallett-Mossop process as the most likely mechanism producing

secondary ice concentrations around −5 °C (Crawford et al., 2012; Taylor et al., 2016; Ladino et al., 2017). Independent of temperature we observed IMFs between $10^1$ and $10^3$ (see Figure 11). At temperatures warmer −15 °C, previous studies consistently confirm this observation and suggest that even IMFs of $10^4$ are frequently observed in the atmosphere. The observations of Stith et al. (2011) and Pasquier et al. (2021) suggest SIP occurring also at temperatures much lower than −15 °C supporting our findings. Especially at these colder temperatures, the contribution of SIP processes remains uncertain





due to the lack of available measurements. At these low temperatures, that are often encountered higher up in the troposphere, our method is a promising tool to investigate IMFs for many clouds and associate ranges of IMFs to regimes of environmental conditions.

## 4 Conclusions and outlook

In this study we retrieved atmospheric INP concentrations in dust-dominated and continental air masses, and IMFs in winter-
time orographic clouds using active remote sensing and in situ observations obtained during the RACLETS field campaign in the Swiss Alps in February and March 2019. Using in situ aerosol and INP observations on a mountaintop in proximity to a lidar located in a nearby high valley, we investigated the remote sensing retrieval performance of aerosol properties and different INP parameterizations for dust and continental aerosol loaded air masses. In addition, we validated the retrievals of ICNC and IWC derived from the combination of radar and lidar data with in situ cloud observations. We proposed a methodol-
ogy to estimate IMFs from collocated lidar and radar data using a tuned INP parameterization for the continental aerosol. We conclude with the following key findings:

1. The aerosol retrieval of the lidar could be validated. We found that the retrieval of particle number concentration $n_{250}$ is less prone to bias than the retrieval of surface area concentration.

2. The dust parameterization by DeMott et al. (2015) in combination with the calibration factor derived by Schrod et al.
(2017) was found to predict the INP concentrations from Saharan dust-carrying air masses best (within a factor ten). Although the calibration factor was derived for condensation and deposition mode INP concentrations at temperatures $\leq -20$ °C and below, we could extend its applicability for immersion mode INP concentrations at temperatures $\geq -20$ °C.

3. We found that the INP parameterization by DeMott et al. (2010) in its original form overpredicts continental INP con-
centration in the Swiss Alps. We propose a multiplicative calibration factor ($\text{cf} = 0.0204$) to significantly improve its performance making it applicable to wintertime continental aerosol.

4. Retrievals of ICNC and IWC from collocated lidar and radar measurements were found to agree with in situ cloud observations within an order of magnitude.

5. We presented a methodology to estimate IMF from collocated lidar and radar measurements. SIP was active nearly 84
% of the time and at all observed temperatures roughly between $-30$ °C and $-5$ °C. Overall, the median IMF found in MPCs over the Swiss Alps was around 80 with an interquartile range of two orders of magnitude. High frequencies of IMF above 80 were observed between $-10$ °C and $-5$ °C based on the restricted number of observed clouds.

6. The IMFs observed with the proposed method showed substantial overlap to IMFs retrieved from previous field obser-vations. Especially at temperatures warmer $-15$ °C, where IMFs between $10^1$ and $10^4$ were frequently observed.





Our study profited from an ideal setting for the closure of continental INP concentration retrieval from the lidar observations combined with in situ observations. The proposed calibration factor should be validated during future field campaigns or on large datasets from past field campaigns with suitable setup. Future development of INP parameterizations intended for lidar applications should be based on aerosol number concentration $n_{250}$ due to the reduced bias in the lidar retrieval. Different assumptions are included in the proposed methodology to determine IMF from collocated lidar and radar observations. How-

ever, the applicability of the methodology is supported by the consistency to the results from previous field observations. The tuning of the D10 parameterization was essential to reduce errors induced by the INP concentration retrieval. As pointed out by Field et al. (2017) reliable quantification of primary ice formation is crucial to constrain SIP. Without the tuning, the INP concentration would have been overestimated by nearly two orders of magnitude and IMFs would have consequently been underestimated by two orders of magnitude. The retrieval of ICNC could be further improved by varying the assumed particle

shapes. Ultimately, the efficiency and occurrence of proposed processes that cause SIP vary based on the environmental conditions (Field et al., 2017; Korolev and Leisner, 2020). It is thus important to link the observed IMFs back to the environmental conditions the cloud formed in. The application of our methodology to large datasets of ground-based or space-borne aerosol and cloud observations could help to assess IMFs in different cloud types and constrain the environmental conditions for different SIP regimes. Machine learning tools such as random forests or gradient boosted decision trees could be valuable tools

in constraining environmental regimes impacting SIP. This could not only accelerate our understanding of ice multiplication in the atmosphere but also be of immense help to motivate and plan laboratory experiments that investigate individual ice multiplication processes.

*Code and data availability.*    Evaluation scripts used in this study are available upon request. The data used in this study will be made publicly accessible on the campaign's website https://www.envidat.ch/group/raclets-field-campaign upon acceptance of the manuscript.





**Appendix A: Categorization of continental background and Saharan dust samples**

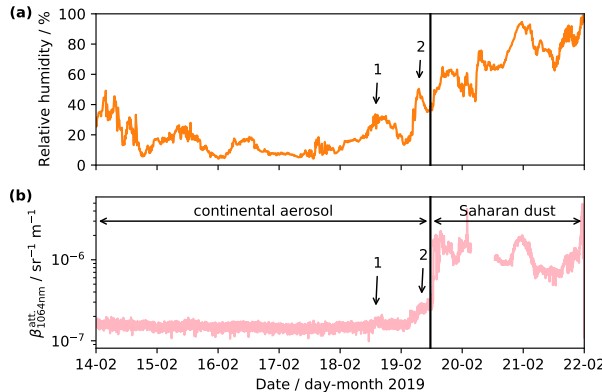

**Figure A1.** (a) relative humidity observed during the comparison period (14–22 February 2019) on 19 February 2019 at WFJ. The two peaks in relative humidity during the transition time are indicated with 1 and 2, respectively. (b) observed lidar attenuated backscatter coefficient $\beta^{\mathrm{att.}}_{1064\mathrm{nm}}$ at the height of WFJ (2687 m a.s.l.). The two characteristic time periods for continental background aerosol and Saharan dust are indicated. Two perturbations in $\beta^{\mathrm{att.}}_{1064\mathrm{nm}}$ during the transition time are indicated with 1 and 2, respectively.

Shortly before the Saharan dust event on 19 February 2019, the Davos region was influenced by air masses already featuring high INP concentrations and a higher aerosol content suggested by an increased signal in the attenuated backscatter (see events 1 and 2 in Figure A1b). The fact that these air masses do not necessarily belong to the following plumes of Saharan dust is further supported by the peaks observed in relative humidity before the phase of Saharan dust (Figure A1a).





 — Atmospheric Chemistry and Physics Discussions, Open Access, EGU

## Appendix B: Overview of individual clouds

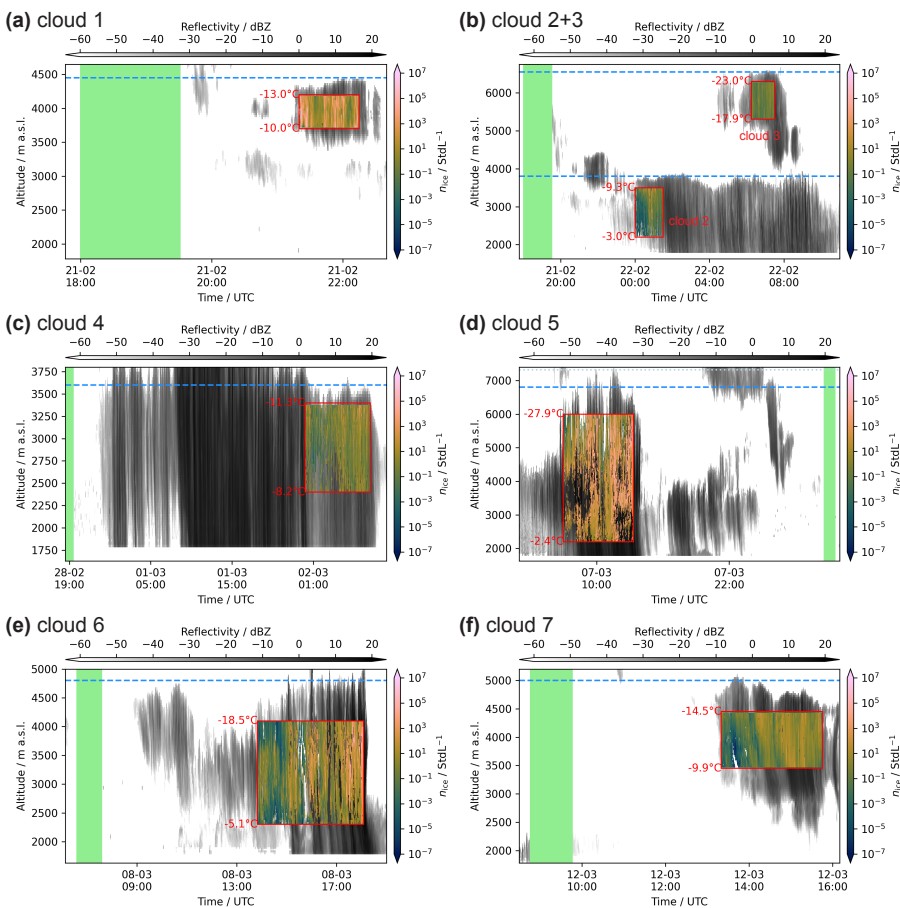

**Figure B1.** As in Figure 9a for each panel except for the parameter labeling of $h_\mathrm{d}$, $t_\mathrm{d}$ and $n_\mathrm{INP,ct}$. Calculated and retrieved parameters for each cloud are presented in Table D1. The blue dotted line in (d) indicates the height above which homogeneous freezing is possible ($T < -38\,°C$).

The exemplary procedure for obtaining IMFs was described for cloud 7 observed on 12 March 2019 in Figure 9. Figure B1 shows the overview for all observed clouds. For cloud 4 (Figure B1c) the last lidar retrieval was more than one day before the ICNC data was available. After the ICNC retrieval time the next lidar profile was also more than a day later. However, the profile used (as indicated in Figure B1c) and the next lidar profile did only differ by a factor of two thus a strong deviation during the available ICNC data times seems unlikely. For cloud 5 the last lidar profile prior the event was only available two days before. Thus, the closest profile in the morning of the next day was used. Small parts of cloud 5 reach temperatures below $-38\,°C$ where potentially homogeneous freezing could occur and additional ice crystals could sediment into higher temperature levels.





## Appendix C: Cloud regime classification

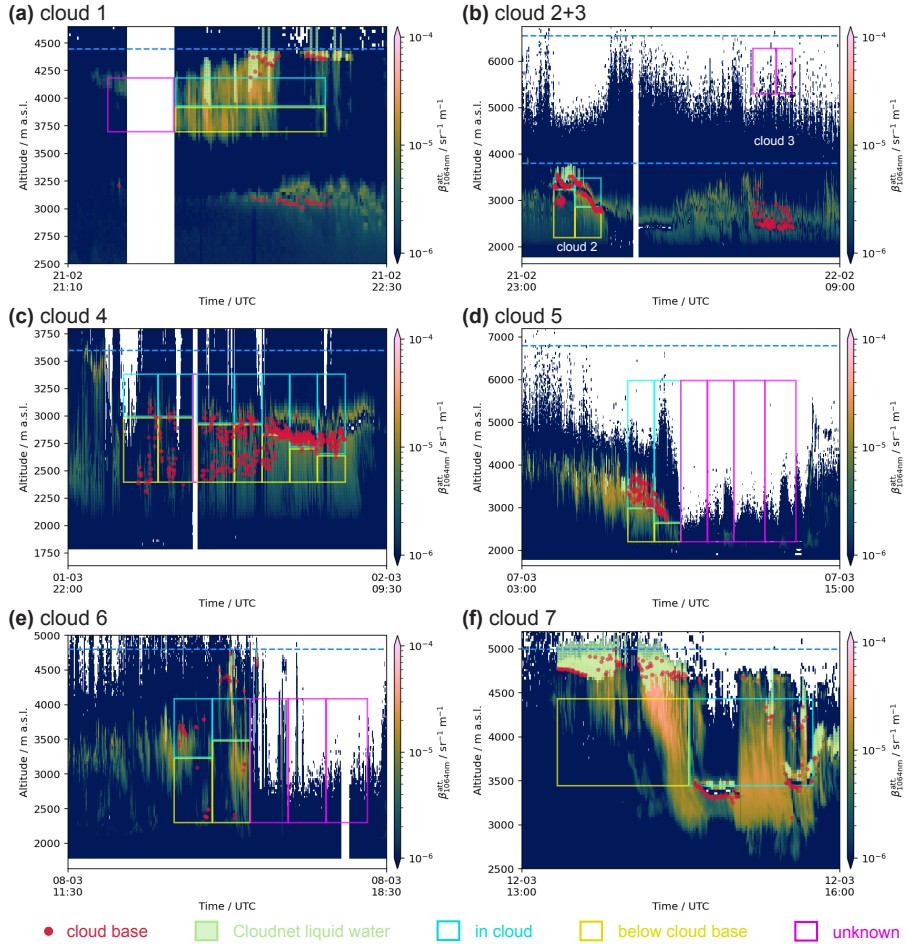

**Figure C1.** Time slicing and classification of the ICNC data indicated in Figure B1 (red rectangle encompasses rectangles presented in this plot). White background represents regions where the lidar was either fully attenuated or the instrument underwent automatic maintenance. The attenuated backscatter coefficient at 1064 nm ($\beta_{1064\mathrm{nm}}^{\mathrm{att.}}$) is presented in the background. A (liquid) cloud base estimate derived from the attenuated backscatter coefficient at 532 nm ($\beta_{532\mathrm{nm}}^{\mathrm{att.}}$) is indicated by red circles. The automated liquid water classification by the Cloudnet algorithm is presented by lime tiles. Empty rectangles in cyan, yellow, and magenta indicate regions classified as above cloud base, below cloud base, and unknown cloud base, respectively. Cloud top derived from radar observations (Figure B1) is indicated by the dashed blue line. The dotted lightblue line in (d) indicates the height above which homogeneous freezing is possible ($T < -38\,°\mathrm{C}$).

The exemplary procedure for obtaining IMFs was described for cloud 7 observed on 12 March 2019 in Figure 9. Figure C1 shows the classification for all observed clouds. Cloud 3 (Figure C1b) was entirely classified as *unknown* due to the liquid water layer at around 2500 m.



**Table D1.** Overview of the SIP analysis per investigated cloud and environmental information for each cloud. For each cloud the total number of ICNC observations ($N_{\text{obs.}}$), the median ICNC ($\widetilde{n}_{\text{ice}}$), the times when SIP was active (SIP active, i.e. percentage of IMFs > 1), the median IMF ($\widetilde{\text{IMF}}$) and its interquartile range ($\text{IQR}_{\text{IMF}}$, in orders of magnitude) are presented for all ICNC data available and ICNC data above a detectable cloud base. Further, the concentration of particles larger 250 nm in radius and INPs at cloud top before the event ($n_{250,\text{ct}}$, $n_{\text{INP,ct}}$), the cloud height ($h_{\text{ct}}$), the height ($h_{\text{d}}$) and time ($t_{\text{d}}$) over which ICNC was obtained, and the temperatures at cloud top ($T_{\text{ct}}$), the top ($T_{\text{dt}}$) and bottom ($T_{\text{db}}$) of available ICNC data are given. In the last column, the total number of observation ($N_{\text{obs.}}$) and averages (weighted by number of observations) of the variables are given combinedly for all clouds.

| | Cloud 1 | Cloud 2 | Cloud 3 | Cloud 4 | Cloud 5 | Cloud 6 | Cloud 7 | all |
|---|---|---|---|---|---|---|---|---|
| **SIP estimate - all ICNC data** | | | | | | | | |
| $N_{\text{obs.}}$ | 1700 | 5900 | 4800 | 16000 | 38000 | 21000 | 9000 | 96000 |
| $\widetilde{n}_{\text{ice}}$ / StdL$^{-1}$ | 26 | 0.032 | 0.22 | 0.63 | 92 | 1.3 | 0.64 | 3.1 |
| SIP active / % | 100 | 72 | 91 | 97 | 91 | 70 | 77 | 85 |
| $\widetilde{\text{IMF}}$ | 5200 | 14 | 10 | 140 | 340 | 21 | 23 | 89 |
| $\text{IQR}_{\text{IMF}}$ / o.o.m. | 2.2 | 2.3 | 0.96 | 1.4 | 2.0 | 2.9 | 2.1 | 2.1 |
| **SIP estimate - ICNC above cloud base** | | | | | | | | |
| $N_{\text{obs.}}$ | 680 | 2400 | 0 | 10000 | 8600 | 4000 | 4400 | 30000 |
| $\widetilde{n}_{\text{ice}}$ / StdL$^{-1}$ | 23 | 0.29 | – | 0.81 | 8.9 | 0.018 | 2.6 | 1.1 |
| SIP active / % | 99 | 87 | – | 67 | 26 | 33 | 96 | 48 |
| $\widetilde{\text{IMF}}$ | 4600 | 130 | – | 180 | 33 | 0.29 | 95 | 78 |
| $\text{IQR}_{\text{IMF}}$ / o.o.m. | 2.3 | 1.1 | – | 1.4 | 2.8 | 2.6 | 1.2 | 2.0 |
| **Cloud information** | | | | | | | | |
| $n_{250}$ / $\cdot10^{-3}$ Stdcm$^{-3}$ | 0.47 | 0.55 | 0.17 | 6.6 | 1.1 | 1.4 | 0.76 | 2.0 |
| $n_{\text{INP,ct}}$ / $\cdot10^{-3}$ StdL$^{-1}$ | 4.9 | 2.2 | 21 | 4.5 | 270 | 62 | 28 | 130 |
| $h_{\text{ct}}$ / km a.s.l. | 4.5 | 3.8 | 6.5 | 3.6 | 6.8 | 4.8 | 5.0 | 5.4 |
| $h_{\text{d}}$ / m | 500 | 1300 | 1000 | 1000 | 3800 | 1800 | 1000 | 2300 |
| $t_{\text{d}}$ / h | 0:55 | 1:30 | 1:15 | 8:00 | 6:20 | 4:15 | 2:25 | 3:31 |
| $T_{\text{ct}}$ / °C | −14 | −10 | −25 | −11 | −33 | −22 | −18 | −23 |
| $T_{\text{dt}}$ / °C | −13 | −9.3 | −23 | −11 | −28 | −19 | −15 | −10 |
| $T_{\text{db}}$ / °C | −10 | −3.0 | −18 | −8.2 | −2.4 | −5.1 | −9.9 | −5.6 |

## Appendix D: IMF summary and additional cloud information

Table D1 provides a summary of obtained IMF for all ICNC data (red rectangles in Figure B1) and using only ICNC data above
cloud base (cyan rectangles in Figure C1) to demonstrate the effect of the selection. Generally, the selection does not change





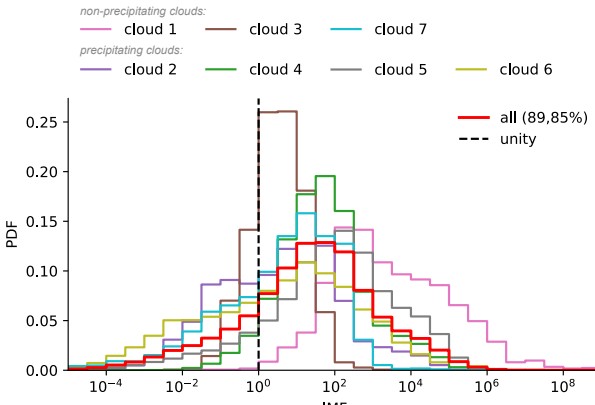

**Figure D1.** Probability density functions (PDFs) of obtained IMFs (Equation 2) for each of the seven observed clouds and all IMFs combined (thick solid red) using all ICNC data (not only above cloud base). The dashed black line indicates unity. The median IMF and the percentage of IMF larger than unity for the combined histogram are given in the legend.

the median IMF (cloud 1) or increases it (clouds 2, 4, and 7). In case of clouds 5 and 6 the median IMF decreased due to the selection. This is explained by the exclusion of high ICNCs that were present during strong precipitation when the lidar was attenuated close to the ground and no cloud base could be determined. Figure D1 shows IMF histograms per cloud and for all clouds combined using all ICNC data.





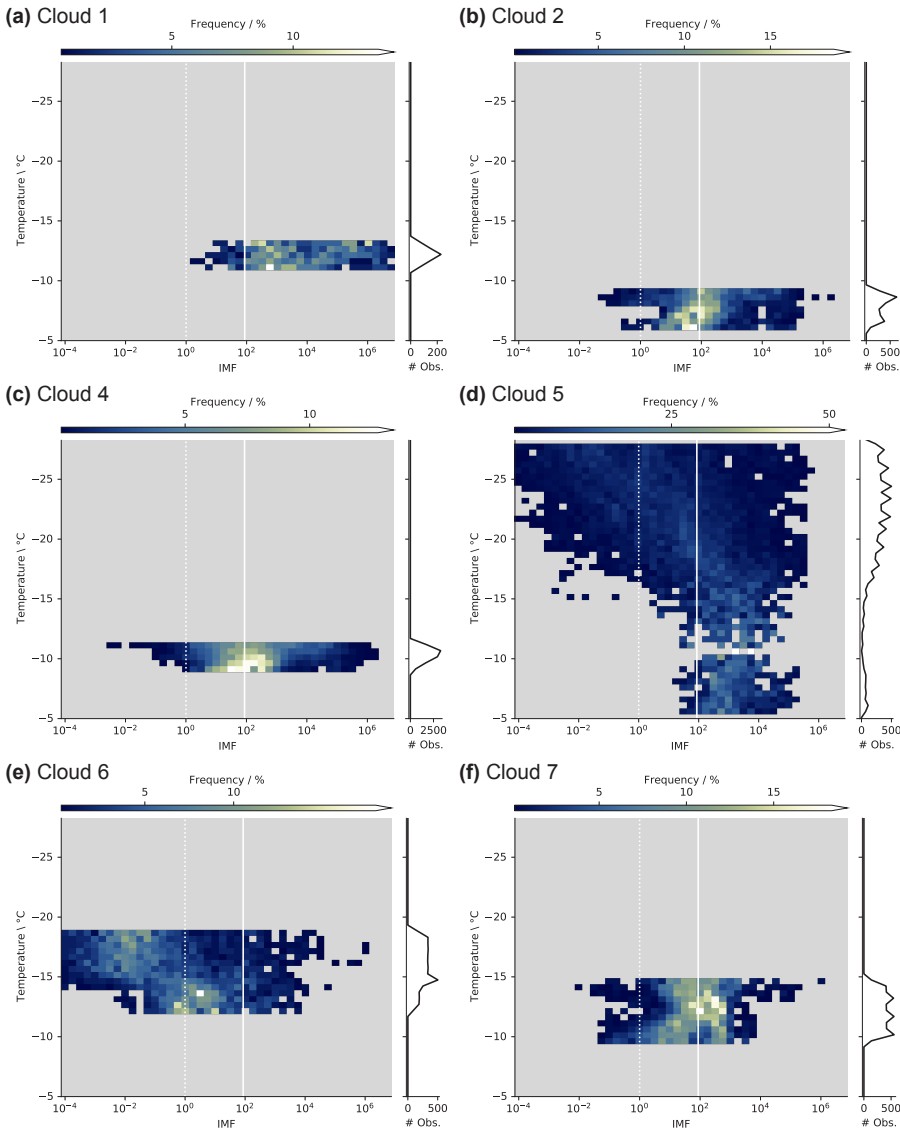

**Figure E1.** Frequency histogram of observed IMFs above cloud base with temperature per cloud. Frequencies are normalized to the number of observations (right axis) per temperature bin. Radar height was converted to temperature using data from the COSMO-1 analysis. Gray background indicates where no data was observed. The dashed white line indicates unity. The median IMF of all data is indicated in solid white.

**Appendix E: IMF temperature histogram per cloud**

IMF histograms with temperature for each cloud for ICNC data above cloud base are presented in Figure E1. Note that for cloud 3 no cloud base could be determined (see Figure C1b).





**Appendix F: Sensitivity of obtained IMF based on data selection**

**Table F1.** Benchmarks of sensitivity studies of obtained SIP and IMF based on data selection for both estimates. The data selection limits in height and time were changed by $\pm$ 200 m and by $\pm$ 20 % of the time span, respectively.

| label | SIP active / % | $\widetilde{\mathrm{IMF}}$ | $\mathrm{IQR_{IMF}}$ / o.o.m. |
|---|---|---|---|
| reference | 85 | 89 | 2.1 |
| $h_{t-200\mathrm{m}}$ | 85 | 92 | 2.1 |
| $h_{b+200\mathrm{m}}$ | 85 | 100 | 2.1 |
| $h_{t-200\mathrm{m},b+200\mathrm{m}}$ | 86 | 110 | 2.1 |
| $t_{s+20\%}$ | 90 | 140 | 1.9 |
| $t_{e-20\%}$ | 82 | 64 | 2.2 |
| $t_{s+20\%,e-20\%}$ | 88 | 110 | 2.0 |

To assess the robustness of our methodology we performed a sensitivity study by changing the upper and lower height of the selected ICNC by 200 m and the starting and ending time by 20 % of the entire time span towards the center (see $h_d$ and $t_d$ in Figure 9a). Note that this test was done on the entire ICNC data available, i.e. the data frames indicated in red in Figure B1. As seen in Table F1 the obtained SIP parameters did not drastically change upon changes in the data selection. The fraction of time when SIP was active did not change by more than five percent points (highest upon changing the starting time by 20 %, $t_{s+20\%}$) and the IMF related parameters did not change by more than a factor of 1.5 (highest upon changing the starting time by 20 %, $t_{s+20\%}$).





**Table G1.** Overview of the studies considered in the IMF comparison and the corresponding cloud type sampled in the study.

| Reference | Cloud type / system | Measurement information | | |
| --- | --- | --- | --- | --- |
| | | platform | ICNC obs. | INP obs. |
| Stith et al. (2011) | Maritime storm | aircraft | in situ | in situ |
| Crawford et al. (2012) | Continental cumuli | aircraft | in situ | literature estimate |
| Lasher-Trapp et al. (2016) | Tropical maritime cumuli | aircraft | in situ | in situ |
| Taylor et al. (2016) | Continental cumuli | aircraft | in situ | in situ aerosol + D10 |
| Ladino et al. (2017) | Tropical cumuli | aircraft | in situ | in situ aerosol + D10 |
| Mignani et al. (2019) | Alpine MPCs | dendrite collection | in situ | in situ |
| Lauber et al. (2021) | Orographic MPC | gondola | in situ | in situ |
| Li et al. (2021) | Stratiform clouds | ground-based | radar | literature estimate |
| Pasquier et al. (2021) | Arctic MPCs | tethered balloon | in situ | in situ |

## Appendix G:  Extraction of IMFs from previous publications

Airborne ice crystal measurements could have overestimated the ICNC due to shattering of ice crystals on the cloud probes (see e.g., Field et al., 2006; Korolev et al., 2011). The retrieval of ICNC from radar observations have also become more reliable with the further technical development of the instruments. Hence, we considered publications from the last decade

to compare our obtained IMF to them (Table G1). However, most studies provide information of ambient temperature, INP concentration, and ICNC separately. Additionally, INP concentrations are typically given at the same location or temperature as the observed ICNC - in contrast to our approach. In the following we briefly describe how IMFs were retrieved from each study. From Stith et al. (2011) Figure 8, maximum and minimum IMF were obtained for the two legs 5:22 and 5:42. The ambient temperature ranges during the legs were taken from Stith et al. (2011) Figure 5. Crawford et al. (2012) estimate an upper limit

of 0.01 L$^{-1}$ in INP concentration based on source considerations. From Crawford et al. (2012) Figure 3, maximal ICNC with temperature were taken. A minimal ICNC could not be distinguished from zero such IMFs smaller than 1 are possible, but not further determinable. Lasher-Trapp et al. (2016) Figure 5 presents the excess of ICNC over INP concentration combinedly as a function of temperature. We considered ICNC data that were not prone to possible side contamination as the authors write. Taylor et al. (2016) Figure 13 provides ICNC with estimates on INP concentration with temperature. Note that we consider

Run 11.1 as point measurement. Ladino et al. (2017) Figure 4 encompasses an extensive overview of their ICNC measurements and predicted INP concentration with temperature. IMFs were calculated using the envelope of all ICNC observations and the INP concentration at corresponding altitude. Mignani et al. (2019) provide IMF between four and 25 (Figure 3) at temperatures between −17 and −12 °C (Figure 2). Lauber et al. (2021) Figure 6 provides a median ICNC concentration close to the melting layer. INP concentrations are below the detection at this temperature are were estimated to be smaller than the lower detection

limit of the drop freezing instrument. For being measurements close to the melting layer a temperature of −0.5 °C was used





for plotting. IMFs from Pasquier et al. (2021) were provided by personal communication. Li et al. (2021) do not provide correlated data of ICNC, INP concentration, and temperature. They state that *ice number concentrations tend to be $1-3$ orders of magnitude higher than expected INP concentrations* for *clouds at temperatures of $-10\ °C$ or warmer*.



*Author contributions.* JW and JH conceived the study. NI performed the initial analysis. JW performed the final analysis and prepared the figures for the manuscript. JW and CM performed the aerosol measurements during the campaign. MH provided the processed lidar data. MH performed manual optimization of the data quality to lower the detection limit of the lidar in the clean environment. JB and PS operated the radar and provided the processed radar data. RE performed the lidar measurements. FR provided the in situ cloud data. JW, NI, CM, MH, ZK, UL and JH interpreted the data. JW wrote the manuscript with contributions from all co-authors. All authors reviewed the manuscript. JH supervised the project.

*Competing interests.* The authors declare no competing interests.

*Acknowledgements.* The authors express their gratitude to the whole RACLETS campaign team for their support and many fruitful discussions. Especially, we thank Michael Lehning (WSL/SLF, EPFL) and his team for their support in realizing the RACLETS campaign. A big thanks to Mario Schär and Lucie Roth for their help with the aerosol measurements. We thank Paul Fopp for providing his land at Wolfgangpass and Martin Genter for logistical support at Weissfluhjoch. Our deepest appreciation to Michael Rösch and Marco Vecellio for technical support. We express our deepest gratitude to Nora Els (University of Innsbruck) for providing us with a second impinger. We thank MeteoSchweiz for the meteorological observations and the COSMO-1 analysis data. We thank Annett Skupin and Hannes Griesche for setting up the lidar at Davos and performing the lidar measurements. The authors thank Franz Conen, Robert David, Julie T. Pasquier, Maxim Samarin and Colin Tully for discussions and suggestions improving the manuscript. Annika Lauber is thanked for providing the photograph of the setup at Wolfgangpass. We acknowledge the use of the scientific color maps provided by Crameri et al. (2020). This study received funding from the Swiss National Science Foundation (grant numbers 200021_169620, 200021_175824).





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
