# Peer review of "Retrieving ice-nucleating particle concentration and ice multiplication factors using active remote sensing validated by in situ observations"

_Atmospheric Chemistry and Physics, 2022_

## Referee Comment (RC1)

**Summary/Recommendations**

"Retrieving ice nucleating particle concentration and ice multiplication factors using active remote sensing validated by in situ observations" is a thorough and well-written manuscript that verifies ice nucleating particle (INP) lidar and radar estimates to in situ measurements to then provide estimates of ice multiplication factors (IMFs). Along the way, the authors carefully work to test different existing parameterizations and offer updates/recommendations based on their findings. Finally, they compare their findings to previous studies (at times estimating IMFs when this information was not provided explicitly in the original work). This paper should be of interest to INP specialists and also to lidar/radar specialists. However, these groups may not have strong expertise in each other's work - for example, an INP specialist may not immediately know the definition of LDR and a lidar/radar specialist may be confused when INP concentrations at a specific temperature are presented. I recommend that the authors take a critical read-through of this manuscript with only their "INP" or "lidar/radar" hat on to find places where brief explanations or reminders could be helpful. I have highlighted a few instances in my general comments below, but strongly encourage the authors to define field-specific terminology throughout. I recommend that this paper be published after addressing these comments.

**General Comments**

I recommend frequently reminding the reader what acronyms stand for. For example, the authors could re-define an acronym in a new section if that acronym doesn't occur in the previous section. I could see readers jumping around between sections so this can ease the burden of keeping track of everything. I also would remind the reader that $n_{ice}$ is a variable name for ICNC and $n_{INP}$ is a variable name for INP throughout (check your figure captions!). Especially be sure to define acronyms in figure captions for readers who are mainly scanning through figures.

Similarly, it could be helpful to remind the reader throughout that WOP is the valley site and that WFJ is the mountaintop site when the distinction is important (e.g. lines 215-219).

Section 2.1 - why was the aerosol line so warm? Wouldn't 46C drive off many semi volatile compounds, leading to a particle size distribution that is different (shifted to smaller sizes) than the ambient SD? Possibly this was characterized/discussed in another of the RACLET papers. I recommend briefly summarizing those findings, if previously discussed, or discussing here. I would suspect that the uncertainties introduced by aerosol evaporation are less than general INP uncertainties but still good to acknowledge, as many of the parameterizations used in this paper rely on particle size.

Section 2.1 - I recommend explicitly splitting out the APS and SMPS size ranges. What did you do for size overlaps, if any? Later on you discuss using only the APS measurements for the D10 parameterization, which relies on particles >500 nm. Was the SMPS completely below that size range?

Line 254 - this is off-line INP at -13 degC, not retrieved INP? Good to make that distinction throughout.

Line 292 - reference the table that D15, H19, U17d are described in

Section 4 - redefine acronyms (e.g. line 529 - INP , IMF)

**Figures/Tables**

I generally enjoyed all of the figures - the colors chosen were pleasing and a large amount of information was summarized well throughout. I will note that the labels on many of the figures were quite small (e.g. figure 2). If the authors feel confident that the figures will be sufficiently large and readable upon publication, I am fine with them being left as is - but redoing some of the labels to be larger may be a big improvement.

Figure 1 - caption has discrepancy between how OCEANET/OceaNet is written.

Table 2 - why is there a 9(11) in the first row? Describe in caption.

Figure 10 - remind reader in the caption that ICNC = $n_{ice}$ (x axis label)

Figure B1 - explain the green patch in each panel.

**Technical comments**

Line 139 - update the second 'APS' to 'SMPS'

Line 240 - missing a word or  phrase, update to something like "Despite t*he fact* that the comparison in this section was purely qualitative…" (italics mine to point out the change)

Line 474 - missing a word or phrase, update to something like "...using the IMF itself does not allow *us* to infer the underlying SIP process…"

Line 540 - update to "(within a factor *of* ten)"

---

## Author Comment (AC1)

Reviewer comments are reproduced in **bold** and author responses in *italic*; extracts from the original manuscript are presented in *red italic*, and from the revised manuscript in *blue italic*.

**"Retrieving ice nucleating particle concentration and ice multiplication factors using active remote sensing validated by in situ observations" is a thorough and well-written manuscript that verifies ice nucleating particle (INP) lidar and radar estimates to in situ measurements to then provide estimates of ice multiplication factors (IMFs). Along the way, the authors carefully work to test different existing parameterizations and offer updates/recommendations based on their findings. Finally, they compare their findings to previous studies (at times estimating IMFs when this information was not provided explicitly in the original work). This paper should be of interest to INP specialists and also to lidar/radar specialists. However, these groups may not have strong expertise in each other's work - for example, an INP specialist may not immediately know the definition of LDR and a lidar/radar specialist may be confused when INP concentrations at a specific temperature are presented. I recommend that the authors take a critical read-through of this manuscript with only their "INP" or "lidar/radar" hat on to find places where brief explanations or reminders could be helpful. I have highlighted a few instances in my general comments below, but strongly encourage the authors to define field-specific terminology throughout. I recommend that this paper be published after addressing these comments.**

*We would like to thank the Referee 1 for reviewing our manuscript. We are pleased with the positive reception and grateful for the helpful comments which improved our manuscript and are answered individually hereafter.*

**I recommend frequently reminding the reader what acronyms stand for. For example, the authors could re-define an acronym in a new section if that acronym doesn't occur in the previous section. I could see readers jumping around between sections so this can ease the burden of keeping track of everything. I also would remind the reader that $n_{ice}$ is a variable name for ICNC and $n_{INP}$ is a variable name for INP throughout (check your figure captions!). Especially be sure to define acronyms in figure captions for readers who are mainly scanning through figures.**

*We thank the reviewer for the suggestion and agree that a manuscript in between two research subfields should offer easy access to the necessary information that belong to one of the specialized fields. We went through the manuscript and added brief explanations where necessary, e.g., in lines 179-181 (revised manuscript)*

*The backscatter contributions are transferred to extinction contributions using an extinction-to-backscatter ratio (lidar ratio) of 55 sr for dust and 50 sr for continental particles (rural background, pollution).*

*and reintroduced abbreviations at nearly 30 instances – especially consistently in captions.*

**Similarly, it could be helpful to remind the reader throughout that WOP is the valley site and that WFJ is the mountaintop site when the distinction is important (e.g. lines 215-219).**

*We thank the reviewer for the suggestion and agree that reminding the reader of the relative locations is beneficial. We added this information in lines 275-277 (revised manuscript) as shown below and at 15 further instances throughout the manuscript.*

*For the comparison of lidar observations to in situ observations at WFJ (mountaintop site) and WOP (high valley site), lidar retrievals were taken from the closest height bins at 2695 m a.s.l. and 2052 m a.s.l., respectively (Figure 1b).*

**Section 2.1 - why was the aerosol line so warm? Wouldn't 46C drive off many semi volatile compounds, leading to a particle size distribution that is different (shifted to smaller sizes) than the ambient SD? Possibly this was characterized/discussed in another of the RACLET papers. I recommend briefly summarizing those findings, if previously discussed, or discussing here. I would suspect that the uncertainties introduced by aerosol evaporation are less than general INP uncertainties but still good to acknowledge, as many of the parameterizations used in this paper rely on particle size.**

*We thank the reviewer pointing out the need for more explanation. We extended the discussion by lines 137-142 (revised manuscript) as follows:*

*At both sites, ambient air was sampled through a 46°C-heated inlet. The heating was a preventive measure to avoid icing of the outside inlet parts, to evaporate activated cloud droplets, and to sublimate ice crystals. The evaporation of volatile compounds of the aerosol cannot fully be excluded. However, the effect is expected to be minor given the high flow rate through the inlet (300 L min$^{-1}$), such that the temperature of sampled air was likely below 46 °C. Furthermore, the degradation of relevant INPs (mostly biological) should only occur at temperatures above 46 °C (Kanji et al., 2017; Huang et al., 2021) and is hence regarded as unlikely (see Wieder et al., 2022b, for further details).*

*Further, we added to the description of the aerosol measurements in lines 151-154 (revised manuscript):*

*As described above, we cannot fully exclude the loss of volatile compounds which could shift the size distributions to smaller sizes, thus underestimating in situ predicted INP concentrations when using size dependent aerosol number concentrations as an input parameter (Section 3.2.3). However, as we discussed above the effect is likely negligible.*

*Lastly, in the discussion of the derivation of the calibration factor for D10, we mention the potential evaporation influence in lines 399-403 (revised manuscript):*

*Furthermore, as discussed in Section 2.1, the use of a heated inlet could potentially lead to the loss of volatile compounds. This could subsequently decrease the obtained $n_{250}$, causing an underestimation of in situ predicted INP concentrations. We note that this effect does not apply to lidar-retrieved aerosol number concentrations and therefore lidar-retrieved INP concentrations. Therefore, the similarity of the performance measures also indicates the validity of the assumption that evaporation of volatile compounds is inconsequential.*

**Section 2.1 - I recommend explicitly splitting out the APS and SMPS size ranges. What did you do for size overlaps, if any? Later on you discuss using only the APS measurements for the D10 parameterization, which relies on particles >500 nm. Was the SMPS completely below that size range?**

*We thank the reviewer for the suggestion and agree that more information about the usage and processing of the size distribution data is very valuable to the reader. We restructured and changed lines 142-151 (revised manuscript) to:*

*Downstream, an Aerodynamic Particle Sizer Spectrometer (APS; Model 3321, TSI Inc., US) and a Scanning Mobility Particle Sizer Spectrometer (SMPS; Model 3938, TSI Inc., US) recorded aerosol size distributions between approximately 10 nm (electric mobility) and 20 µm (aerodynamic diameter). In this study, electric mobility (SMPS) and aerodynamic diameter (APS) were converted to physical diameter assuming a shape factor χ = 1.2 and assuming a particle density ρ = 2 g cm$^{-3}$ (Thomas and Charvet, 2017). After conversion, the observed size range of the SMPS and the APS covered particles with physical diameters between 10 nm and 400 nm, and between 400 nm and 15 µm, respectively. The total surface area concentration (hereafter referred to as s) of all aerosols was calculated from the entire size distribution utilizing both SMPS and APS data. The number concentrations of particles with radii ≥ 250 nm (hereafter referred to as $n_{250}$) was obtained by integrating the APS distributions for physical diameters ≥ 500 nm.*

**Line 254 - this is off-line INP at -13 degC, not retrieved INP? Good to make that distinction throughout.**

*Yes, thank you. We added the information at some instances such as in lines 311-313 (revised manuscript):*

*INP concentrations at −13 °C measured in situ at WFJ also showed an increase of approximately one order of magnitude already on 18 February (a day before the strong Saharan dust signal in Figure 3a and 3b).*

**Line 292 - reference the table that D15, H19, U17d are described in**

*We changed lines 349-352 (revised manuscript) to:*

*In Figure 4, we present comparisons to the parameterizations of D15, H19, and U17d (Table 1) along with performance measures […]*

*Further, we also referenced the table in the consequent subsection changing lines 376-379 (revised manuscript) to:*

*In Figure 5, we compare the in situ observed INP concentration of these samples with lidar retrievals using the parameterizations: (a) D10 for global aerosol and (b) U17s for soot which are the best parameterizations available to mimic continental aerosol (Table 1).*

**Section 4 - redefine acronyms (e.g. line 529 - INP , IMF)**

*We redefined all acronyms in the conclusions.*

**I generally enjoyed all of the figures - the colors chosen were pleasing and a large amount of information was summarized well throughout. I will note that the labels on many of the figures were quite small (e.g. figure 2). If the authors feel confident that the figures will be sufficiently large and readable upon publication, I am fine with them being left as is - but redoing some of the labels to be larger may be a big improvement.**

*We thank the reviewer for the overall positive reception of the presented figures. Following the suggestion to improve the figure readability we enlarged the label sizes of most of the figures in the manuscript.*

**Figure 1 - caption has discrepancy between how OCEANET/OceaNet is written.**

*We thank the reviewer for spotting the discrepancy and updated the caption of Figure 1 accordingly.*

*[…] The OCEANET container and the lower aerosol monitoring site were located at Wolfgangpass (blue dot, WOP, 1631 m a.s.l.). […]*

**Table 2 - why is there a 9(11) in the first row? Describe in caption.**

*We thank the reviewer for pointing out the need for more clarity. We added a footnote to Table 2 and changed the caption of Table 2 to:*

**Table 2.** *Overview of the number of samples (N) available for INP-closure in dust-dominated and continental air mass cases. Additionally, the number of days over which the samples are taken is given.*

| Air mass | $N$ | days |
|---|---|---|
| dust | 9 (11*) | 3 |
| continental | 23 | 14 |

\* Note that due to technical maintenance of the lidar instrument only nine of the 11 total samples during the Saharan dust event shown in Figure 3 could be used.

**Figure 10 - remind reader in the caption that ICNC = $n_{ice}$ (x axis label)**

*We changed the caption of Figure 10 to:*

*Median ice crystal number concentrations (ICNCs) as a function of altitude for each cloud and for time slices of approximately one hour. (a) Temporal evolution of the ICNC ($n_{ice}$) profiles: Faintest colors correspond to the first slice of data and the strongest colors correspond to the last slice of data. (b) Cloud regime classification for each time slice: ICNCs ($n_{ice}$) taken above cloud base (solid), below cloud base (dashed) and unknown cloud base (faint color). For cloud 3, the cloud base could not be determined.*

**Figure B1 - explain the green patch in each panel.**

*We thank the reviewer for pointing out the need for more clarity. We changed the caption of Figure B1 to:*

*As in Figure 9a for each panel except for the parameter labeling of $h_d$, $t_d$ and $n_{INP,ct}$. The green shading indicates the period during which the INP concentration at cloud top (dashed blue line) was retrieved. Calculated and retrieved parameters for each cloud are presented in Table D1. [...]*

**Line 139 - update the second 'APS' to 'SMPS'**

*Thanks, we updated accordingly.*

**Line 240 - missing a word or phrase, update to something like "Despite *the fact that* the comparison in this section was purely qualitative..." (italics mine to point out the change)**

*Thanks, we changed line 297 (revised manuscript) to:*

*Despite the fact that the comparison in this section was purely qualitative, two important insights are gained:*

**Line 474 - missing a word or phrase, update to something like "...using the IMF itself does not allow *us* to infer the underlying SIP process..."**

*We changed lines 538-539 (revised manuscript) to:*

*As stated in the introduction, using the IMF (as defined in Equation 2) itself does not allow the inference of the underlying SIP process due to the potential (temporal and spatial) displacement of first ice crystal formation and occurrence of SIP.*

**Line 540 - update to "(within a factor *of* ten)"**

*We changed lines 605-606 (revised manuscript) to:*

*The dust parameterization by DeMott et al. (2015) in combination with the calibration factor derived by Schrod et al. (2017) was found to predict the INP concentrations from Saharan dust-carrying air masses best (within a factor of ten).*

**References**

DeMott et al. (2015), Huang et al. (2021), Kanji et al. (2017), and Schrod et al. (2017) as in manuscript.

Wieder, J., Mignani, C., Schär, M., Roth, L., Sprenger, M., Henneberger, J., Lohmann, U., Brunner, C., and Kanji, Z. A.: Unveiling atmospheric transport and mixing mechanisms of ice-nucleating particles over the Alps, Atmospheric Chemistry and Physics, 22, 3111–3130, https://doi.org/10.5194/acp-22-3111-2022, https://acp.copernicus.org/articles/22/3111/2022/, 2022b.

---

## Author Comment (AC2)

Reviewer comments are reproduced in **bold** and author responses in *italic*; extracts from the original manuscript are presented in *red italic*, and from the revised manuscript in *blue italic*.

**Review for Wieder et al., "Retrieving ice nucleating particle concentration and ice multiplication factors using active remote sensing validated by in situ observations", submitted to Atmospheric Chemistry & Physics**

**This manuscript presents a method to retrieve ice-nucleating particle (INP) concentrations using a polarization Raman lidar and a Ka-band cloud radar as well as ice multiplication factors due to secondary ice production in orographic mixed-phase clouds. The retrievals were compared against in situ observations derived from a tethered balloon system at two locations: the WFJ and WOP sites at different altitudes during the RACELETS field campaign in the Swiss Alps.**
**Retrievals of INP concentrations and ice multiplication are both extremely poorly constrained and valuable for better constraining cloud properties. The study is thus well-motivated and using tethered balloons for ice crystal number concentration is advantageous compared to aircraft in situ probe measurements because they don't suffer from ice crystal shattering effects. However, details describing the methodology and a clear disclosure of assumptions and quantification of limitations is lacking. Specific comments follow.**

*We would like to thank the Referee 2 for reviewing our manuscript. We are pleased with the positive reception and grateful for the helpful comments which improved our manuscript and are answered individually hereafter.*

**The method is not described in sufficient detail. This is especially important given the large number of assumptions that need to be made in the retrievals. For example, the method to retrieve the INP number concentration is appears to use the various INP concentrations, but not a single equation for any of the parameterizations appears in the manuscript. The same goes for the ICNC retrieved by the radar under the assumption of a particular ice crystal size distribution which was not described.**

*We thank the reviewer for pointing out the need for more details about the used methods. We extended the methods section accordingly, especially by providing more details on the lidar retrieval of aerosol properties (Section 2.2), the retrieval of ice crystal properties from both remote sensing (Section 2.3) and the in situ instrument (Section 2.4), and explicit equations of all used INP parameterizations (Section 2.5). Due to the large number of changes please refer to the updated track changes version of the manuscript.*

**Error quantification is almost nonexistent in this work.**

*We thank the reviewer for pointing out the need for more error quantification as we have mostly put an emphasis on the discussion of the induced methodological uncertainties in our manuscript. We provided the errors of all used quantities in the method section (Sections 2.1, 2.2, and 2.3) and indicated them at meaningful positions in different plots, see Figures 2, 3, 4, 5, 6, 7, and 8 in the revised manuscript.*

**Why does the lidar almost always overestimate the in situ observations at the WFJ site? I couldn't find an explanation for this.**

*We thank the reviewer for pointing out the need for clarification. The remote sensing measurements in the very clean air above the Alps are challenging. The previously used conversion factors from the lidar to aerosol concentrations (see Equations 2 and 3 in the revised manuscript) were partially responsible for an overestimation of the aerosol concentrations. We derived new conversion factors from the dataset collected during*

*our campaign which reduced the retrieved concentrations by up to 20 %. Furthermore, given the very low extinction coefficients in the clean air over the Alps, the assumed linear relationship by Mamouri and Ansmann (2016) in Equations 2 and 3 may not hold anymore. An overestimation in the remote sensing observations has also previously been reported by Haarig et al. (2019) using the similar instrument in a Saharan dust layer at Barbados. We extended the argumentation in lines 285-296 (revised manuscript) to:*

*At relatively low aerosol concentrations and in the presence of continental aerosol, a plateauing of the lidar-retrieved aerosol concentrations was observed. The clear atmosphere over the Alps with very low values of the extinction coefficient (< 10 Mm⁻¹) could be responsible for deviations from the assumed linear relationship of extinction to $n_{250}$ and s, respectively, (Mamouri and Ansmann, 2016). The larger diurnal variability between the in situ observation and lidar retrieval at WOP (high valley site) compared to WFJ (mountaintop site) can be explained by the diurnal changes of aerosol concentration near the ground (in situ observations) not affecting the air masses on the lidar retrieval height (height difference approx. 400 m, see Figure 1b). This difference in height and therefore air mass commonly limits a quantitative conclusion between ground-based in situ observations and remote sensing instruments as the well-mixed boundary layer could at times not extend up to the lowest retrieval height. For the retrieval of s (Figures 2c and 2d) the aforementioned observations hold equally true, which is not surprising as the surface area relates to the square of particle radius. However, comparing the retrieval accuracy at WFJ (Figures 2a and 2c), a stronger bias of the lidar retrieved surface area concentrations is apparent which has been also previously reported by Haarig et al. (2019), based on similar observations carried out at Barbados.*

**For the non-expert in in situ measurements, why does the ambient air need to be heated to 46C in the inlet?**

*We thank the reviewer pointing out the need for more explanation. We extended the discussion by lines 137-142 (revised manuscript) as follows:*

*At both sites, ambient air was sampled through a 46°C-heated inlet. The heating was a preventive measure to avoid icing of the outside inlet parts, to evaporate activated cloud droplets, and to sublimate ice crystals. The evaporation of volatile compounds of the aerosol cannot fully be excluded. However, the effect is expected to be minor given the high flow rate through the inlet (300 L min⁻¹), such that the temperature of sampled air was likely below 46 °C. Furthermore, the degradation of relevant INPs (mostly biological) should only occur at temperatures above 46 °C (Kanji et al., 2017; Huang et al., 2021) and is hence regarded as unlikely (see Wieder et al., 2022b, for further details).*

**References**

Haarig et al. (2019), Huang et al. (2021), Kanji et al. (2017), and Mamouri and Ansmann (2016) as in manuscript.

Wieder, J., Mignani, C., Schär, M., Roth, L., Sprenger, M., Henneberger, J., Lohmann, U., Brunner, C., and Kanji, Z. A.: Unveiling atmospheric transport and mixing mechanisms of ice-nucleating particles over the Alps, Atmospheric Chemistry and Physics, 22, 3111–3130, https://doi.org/10.5194/acp-22-3111-2022, https://acp.copernicus.org/articles/22/3111/2022/, 2022b.